# In the Eye of the Beholder:
# Robust Prediction with Causal User Modeling

**Amir Feder**
Columbia University
amir.feder@columbia.edu

**Guy Horowitz**
Technion

**Yoav Wald**
Johns Hopkins University

**Roi Reichart**
Technion

**Nir Rosenfeld**
Technion

## Abstract

Accurately predicting the relevance of items to users is crucial to the success of many social platforms. Conventional approaches train models on logged historical data; but recommendation systems, media services, and online marketplaces all exhibit a constant influx of new content—making relevancy a moving target, to which standard predictive models are not robust. In this paper, we propose a learning framework for relevance prediction that is robust to changes in the data distribution. Our key observation is that robustness can be obtained by accounting for *how users causally perceive the environment*. We model users as boundedly-rational decision makers whose causal beliefs are encoded by a causal graph, and show how minimal information regarding the graph can be used to contend with distributional changes. Experiments in multiple settings demonstrate the effectiveness of our approach.

## 1 Introduction

Across a multitude of domains and applications, machine learning has become imperative for guiding human users in many of the decisions they make [44,58,10]. From recommendation systems and search engines to e-commerce platforms and online marketplaces, learned models are regularly used to filter content, rank items, and display select information—all with the primary intent of helping users choose items that are relevant to them. The predominant approach for learning in these tasks is to train models to accurately predict the relevance of items to users. But since training is often carried out on logged historical records, even highly-accurate models remain calibrated to the distribution of *previously* observed data on which they were trained [7,59,56]. Given that in virtually any online platform the distribution of content naturally varies over time and location—due to trends and fashions, innovation, or forces of supply and demand—models trained on logged data may fail to correctly predict the choices and preferences of users on unseen, future distributions [12,1,18,27,34].

In this paper, we present a novel conceptual framework for learning predictive models of user-item relevance that are robust to changes in the underlying data distribution. Our approach is built around two key observations: (i) that relevance to users is determined by the way in which users *perceive* value, and (ii) that this process of value attribution is *causal* in nature. As an example, consider a video streaming service in which a user $u$ is trying to determine whether watching a certain movie will be worthwhile. To make this decision, $y \in \{0, 1\}$, the user has at her disposal a feature description of the movie, $x$, and a system-generated, personalized relevance score, $r$ (e.g., "a 92% match!"). How will she integrate these two informational sources into a decision? We argue that this crucially hinges on her belief as to *why* a particular relevance score is coupled to a particular movie. For example, if a movie boasts a high relevance score, then she might suppose this score was given *because* the system believes the user would like this movie. Another user, however, may reason differently, and

36th Conference on Neural Information Processing Systems (NeurIPS 2022).

instead believe that high relevance scores are given *because* movies are sponsored; if she suspects this to be a likely scenario, her reasoning should have a stark effect on her choices. In both cases above, perceived values (and the actions that follow) stem from how each user causally interprets the recommendation environment $e$, and the underlying causal structure determines how belief regarding value changes, or does not, in response to changes in important variables (e.g., in $u$, $x$, or $r$).

Here, we show how knowledge regarding the causal perceptions of users can be leveraged for providing distributional robustness in learning. A primary concern for robust learning is the reliance of predictions on spurious correlations [3]; here we argue that spuriousness can *result* from causal perceptions underlying user choice behavior. To see the relation between causal perceptions and spuriousness, assume that in our movies example above, the training data exhibits a strong correlation between users' choices of movies, $y$, and a 'genre' feature, $x_g$. A predictive model optimized for accuracy will likely exploit this association, and rely on $x_g$ for prediction. Now, further assume that what *realy* drives user satisfaction is 'production quality', $x_q$; if $x_g$ and $x_q$ are *spuriously* correlated in the training data, then once the distribution of genres naturally changes over time, the predictive model can fail: the association between $x_g$ and $y$, on which predictions rely, may no longer hold.

In essence, our approach casts robust prediction of personalized relevance as a problem of out-of-distribution (OOD) learning, but carefully tailored to settings where data generation is governed by users' causal beliefs and corresponding behavior. There is a growing recognition of how a causal understanding of the learning environment can improve cross-domain generalization [3,55]; our key conceptual contribution is the observation that, in relevance prediction, users' perceptions *are* the causal environment. Thus, there is no 'true' causal graph—all is in the eye of the beholder. To cope with this, we model users as reasoning about decisions through a causal graph [35,47]—thus allowing our approach to anticipate how changes in the data translate to changes in user behavior. Building on this idea, as well as on recent advances in the use of causal modeling for out-of-distribution learning [3,55,54], we show how various levels of knowledge regarding users' causal beliefs—whether inferred or assumed—can be utilized for learning distributionally robust predictive models.

To encourage predictions $\hat{y}$ to be invariant to changes in the recommendation environment $e$, our approach enforces independence between $\hat{y}$ and $e$ (possibly given $y$). This is achieved through regularization, which penalizes predictive models for relying on $e$ [22]. In general, different graphs require different regularization schemes [54]; in our context, this would seem to imply that invariant learning requires precise information regarding each user's causal graph. However, our key observation is that, for user graphs, it suffices to know which of two *classes* the graph belongs to—*causal* or *anti-causal*— determined by the direction of edges between $x, r$ and $y$ (Fig. 1a). Thus, correctly determining which regularization to apply requires only minimal graph knowledge.

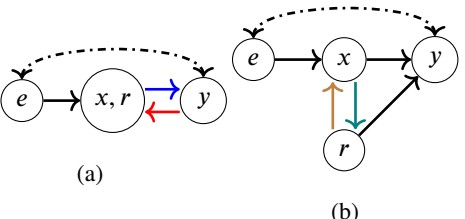

(a)

(b)

Figure 1: Simplified graphs describing users of different: (a) *classes*: *causal* or *anti-causal*, and (b) *subclasses*: *believer* or *skeptic* (here shown for a causal user). Dashed lines indicate possible spuriousness (e.g., via selection).

Nonetheless, more fine-grained information can still be useful. We show the following novel result: if two users generate the same data, but differ in their underlying graph, they will have different optimal *out-of-distribution* predictive models (despite sharing the same optimal in-distribution model). The reason for this is that, to achieve robustness, regularizing for independence will result in the discarding of different information for each user. Operationally, this means that learning should include different models for each user-type (not doing so implicitly constrains the models to be the same). Here again we show that minimal additional information is useful, and focus on subclasses of graphs that differ only in the direction of the edges between $x$ and $r$, which give rise to two user subclasses: *believers* and *skeptics* (Fig. 1b). Nonetheless, our result on differing optimal models applies more broadly, and may be of general interest for causal learning.

We end with a thorough empirical evaluation of our approach (§5), where we explore the benefits of different forms of knowledge regarding users' causal beliefs: whether they are *casual* or *anti-causal*, and whether they are *believers* or *skeptics*. Our results show that learning in a way that accounts for users' causal perception has significant advantages on out-of-distribution tasks. We also study

the degree to which imprecise graph knowledge is useful; our results here show that even a rough estimate of a user's class is sufficient for improved performance, suggesting that our approach can be effectively applied on the basis of domain knowledge or reasonable prior beliefs. Conversely, our results also imply that *not* accounting for causal aspects of user decision-making, or modelling them wrongly, can result in poor out-of-distribution performance. As systems often also play a role in determining what information is presented to users (e.g., providing $r$), understanding possible failure modes—and how to obtain robustness—becomes vital.

**Broader aims.** We aim to promote within machine learning the idea of modeling users as active and autonomous decision-makers, with emphasis on capturing realistic aspects of human decision-making. Our approach crucially hinges on modelling users as decision-makers that (i) reason *causally*, (ii) are *boundedly-rational*, and (iii) must cope with *uncertainty*. As we will show, all three are key to our framework, and operate in unison. Each of the aspects above relates to one of three main pillars on which modern theories of decision-making stand [46], thus blending three different fields—discrete choice (economics), causal modeling (statistics), and domain generalization (machine learning).

## 2  Related Work

**Causality and Recommendations.** Formal causal inference techniques have been used extensively in many domains, but have only recently been applied to recommendations [29,57,7,59,56]. Liang et al. [30] use causal analysis to describe a model of user exposure to items. Some work has also been done to understand the causal impact of these systems on behavior by finding natural experiments in observational data [43,50,40], and through simulations [11,39]. Bottou et al. [8] use causally-motivated techniques in the design of deployed learning systems for ad placement to avoid confounding. As most of this literature addresses selection bias and the effect of recommendations on user behavior [7,59,56], there is no work, as far as we know, that models boundedly rational agents interacting with a recommender system. Moreover, we are the first to propose modeling users' (mis)perceptions about the recommendation generation process using causal graphs.

**Bounded Rationality and Subjective Beliefs.** The bounded rationality literature focuses on modelling agents that make decisions under uncertainty, without the ability to fully process the state of the world, and therefore hold subjective beliefs about the data-generating process. Eyster and Rabin [17] defined *cursed beliefs*, which capture an agent's failure to realize that his opponents' behavior depends on factors beyond those he is informed of. Building on Esponda and Pouzo [16], who modelled equilibrium beliefs under misspecified subjective models, Spiegler [47] used causal graphs to analyze agents that impose subjective causal interpretations on observed correlations. This work lays the foundation upon which we model users here, and has sprouted many interesting extensions [14,15].

**Causality and Invariant Learning.** Correlational predictive models can be untrustworthy [24], and latch onto spurious correlations, leading to errors in OOD settings [33,20,19]. This shortcoming can potentially be addressed by a causal perspective, as knowledge of the causal relationship between observations and labels can be used to mitigate predictor reliance on them [9,54]. In our experiments, we learn a representation that is invariant to interventions on the 'environment' $e$, a special case of an invariant representation [3,28,4]. Learning models which generalize OOD is a fruitful area of research with many recent developments [31,23,36,51,5,55]. Recently, Veitch et al. [54] showed that the means and implications of invariant learning depend on the data's true causal structure. Specifically, distinct causal structures require distinct regularization schemes to induce invariance.

## 3  Modelling Approach

### 3.1  Learning Setting

In our setting, data consists of users, items, and choices. Users are described by features $u \in \mathbb{R}^{d_u}$, and items are described by two types of features: intrinsic item properties, $x \in \mathbb{R}^{d_x}$ (e.g., movie genre, plot synopsis, cast and crew), and information provided by the platform, $r \in \mathbb{R}^{d_r}$ (e.g., recommendation score, user reviews). We will sometimes make a distinction between features that are available to users, and those that are not; in such cases, we denote unobserved features by $\bar{x}$, and with slight abuse of notation, use $x$ for the remaining observed features (we assume $r$ is always observed). Choices $y \in \{0, 1\}$ indicate whether a user $u$ chose to interact (e.g., click, buy, watch) with a certain item $(x, r)$. Tuples $(u, x, r, y)$ are sampled iid from certain unknown joint distributions, which we define next.

As we are interested in robustness to distributional change, we follow the general setup of *domain generalization* [6,26,5,55] in which there is a collection of environments, denoted by a set $\mathcal{E}$, and each environment $e \in \mathcal{E}$ defines a different joint distribution $D^e$ over $(u, x, r, y)$. We assume there is training data available from a subset of $K$ environments, $\mathcal{E}_{\text{train}} = \{e_1, \ldots, e_K\} \subset \mathcal{E}$, with datasets $S_k = \{(u_{ki}, x_{ki}, r_{ki}, y_{ki})\}_{i=1}^{m_k}$ drawn i.i.d from the corresponding $D^{e_k}$. We denote the pooled training distribution by $D_{\text{train}} = \cup_{e \in \mathcal{E}_{\text{train}}} D^e$ and the pooled training data by $S = \cup_k S_k$ with $m = \sum_k m_k$.

Our goal is to learn a robust predictive model $\hat{y} = f(u, x, r; \theta) := f_u(x, r; \theta)$ with parameters $\theta$; the $f_u$ notation will be helpful in our discussion of robustness as it emphasizes our focus on individual users. We now turn to define the precise type of robustness that we will be seeking.

**Robustness via causal graphs.** The type of robustness that we would like our model to satisfy is *counterfactual invariance* (CI) [54]. Denoting $x(e), r(e)$ as the counterfactual features that would have been observed had the environment been set to $e$, this is defined as:

**Definition 1.** *A model $f_u$ is CI if $\forall e, e' \in \mathcal{E}$ it holds a.e. that $f_u(x(e'), r(e'); \theta) = f_u(x(e), r(e); \theta)$.*

The challenge in obtaining CI predictors is that at train time we only observe a subset of the environments, $\mathcal{E}_{\text{train}} \subset \mathcal{E}$, while CI requires independence to hold for *all* environments $e \in \mathcal{E}$. To reason formally about the role of $e$ in the data generating process, and hence about the type of distribution shifts under which our model should remain invariant, it is common to assume that a causal structure underlies data generation [23,3]. This is often modeled as a (directed) causal graph [35]; robustness is then defined as insensitivity of the predictive model to changes (or 'interventions') in the variable $e$, which can trigger changes in other variables that lie 'downstream' in the graph. To encourage robustness, a common approach is to construct a learning objective that avoids spurious correlations by enforcing certain conditional independence relations to hold, e.g., via regularization (see §4). The question of *which* relations are required can be answered by examining the graph and the conditional independencies it encodes (between $e, x, r, y$, and $\hat{y}$). Unfortunately, inferring the causal graph is in general hard; however, determining the 'correct' learning objective may require only partial information regarding the graph. We will return to the type of information we require for our purposes, and the precise ways in which we use it, in §4.

### 3.2 Users as decision makers

Focusing on relevance prediction, at the heart of our approach lies the observation that what underlies the generating process of data, and in particular of labels, *is the way in which users causally perceive the environment*. In this sense, users *are* the causal mechanism, and their causal perceptions manifest in their decision-making. Operationally, we model users as acting on the basis of *individualized causal graphs* [47] that define how changes in one variable propagate to influence others, and ultimately—determine choice behavior. This allows us to anticipate, target, and account for sources of spuriousness.

**Rational users.** To see how modeling users as causal decision-makers can be helpful, consider first a conventional 'correlative' approach for training $f_u$, e.g. by minimizing the loss of a corresponding score function $v_u(x, r) = v(u, x, r)$, and predicting via $\hat{y} = \text{argmax}_{y \in \{0,1\}} y v_u(x, r) = \mathbb{1}\{v_u(x, r) > 0\}$. From the perspective of user modeling, $v_u$ can be interpreted as a personalized 'value function'; this complies with classic Expected Utility Theory (EUT) [53], in which users are modeled as rational agents acting to maximize (expected) value, and under *full information*.[1] From a causal perspective, this approach is equivalent to assuming a graph in which all paths from $e$ to $y$ are blocked by $x, r$—which is akin to assuming no spurious pathways, and so handicaps the ability to avoid them.

**Boundedly-rational users.** We propose to model users as boundedly-rational decision-makers, under the key assertion that users' decisions take place under inherent *uncertainty*. Uncertainty plays a key role in how we, as humans, decide: our actions follow not only from what we know, but also from how we account for what we don't know. Nonetheless, and despite being central to most modern theories of decision making [25]—and despite being a primary reason for why users turn to online informational services like recommendation systems in the first place—explicit modeling of user-side uncertainty is currently rare within machine learning [37,2,38]; here we advocate for its use.

Our modeling approach acknowledges that users *know* some features are unobserved, and that this influences their actions. Here we demonstrate how this relates to robust learning through an illustrative

---

[1]For simplicity, here and throughout we consider deterministic valuations, although this is not necessary.

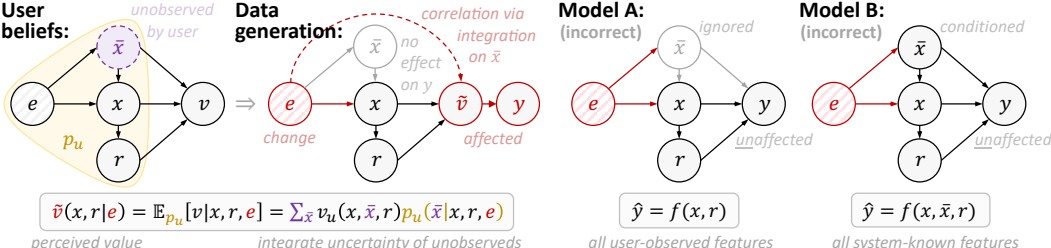

Figure 2: **(Left)** User beliefs and resulting data generation process of $y$. The user does not observe $\bar{x}$, but knows it is missing. To compensate for this uncertainty, the user integrates over $\bar{x}$ w.r.t. probabilistic beliefs $p_u$, which can depend on $e$; this forms her perceived value $\tilde{v}$, which determines her choice $y$. Integrating uncertainty can introduce correlation between $e$ and $\tilde{v}$ (and hence $y$) through $p_u(\cdot|e)$. Note $y$ does not depend on *instances* of $\bar{x}$. **(Center)** Learning a predictive model $f(x,r)$ using only features observed by the user. Assuming that $\bar{x}$ can be discarded creates the impression that information cannot flow from $e$ to $y$, implying (wrongly) that a naïvely trained $f(x,r)$ would be robust. In practice, such an $f$ might incorrectly use $x,r$ to compensate for variation in $e$. **(Right)** Learning a predictive model $f(x,\bar{x},r)$ using all features available to the system. Assuming $\bar{x}$ affects $y$ implies (wrongly) that all paths from $e$ to $y$ are blocked. Naïvely training $f$ will likely use variation in $\bar{x}$ to explain $y$; this might improve performance on observed $e$, but will not generalize to others.

example using a particular behavioral model, though as we will show, our approach applies more broadly. Consider a user shopping online for a vintage coat, and considering whether to buy a certain coat. The coat's description includes several intrinsic properties $x$ (e.g., the coat's material), as well as certain platform-selected information $r$ (e.g., a stylized photo of a vintage-looking coat). The user wants to make an informed decision, but knows some important information, $\bar{x}$, is missing (e.g., the year in which the coat was manufactured). If she is concerned about buying a modern knockoff (rather than a truly vintage coat), how should she act? A common approach is to extend EUT to support uncertainty by modelling users as integrating subjective beliefs about unobserved variables, $p_u(\bar{x}|x,r,e)$, into a conditional estimate of value, $\tilde{v}_u(x,r|e)$, over which choices $y$ are made:

$$\tilde{v}_u(x,r|e) = \sum_{\bar{x}} v_u(x,\bar{x},r)p_u(\bar{x}|x,r,e), \qquad y = \mathbb{1}\{\tilde{v}_u(x,r|e) > 0\} \qquad (1)$$

Here, $p_u(\bar{x}|x,r,e)$ describes a user's (probabilistic) belief regarding the conditional likelihood of each $\bar{x}$ (is the vintage-looking coat truly from the 60's?), and $v_u(x,\bar{x},r)$ describes the item's value to the user *given* $\bar{x}$ (if the coat really is from the 60's—how much is it worth to me?). Importantly, note that uncertainty beliefs $p_u(\bar{x}|x,r,e)$ can be environment-specific (i.e., the degree of suspicion regarding knockoffs can vary across retailers). In turn, value estimates $\tilde{v}$ and choices $y$ can also rely on $e$ (note that $\tilde{v}$ can also be interpreted as the conditional expected value, $\tilde{v}_u(x,r|e) = \mathbb{E}_{p_u(\cdot|e)}[v|x,r]$). Since $y$ is a deterministic function of $\tilde{v}$ (Eq. (1)), for clarity (and when clear from context) we will "skip" $\tilde{v}$ and refer to $y$ as a direct function of $x,r$, and $e$.

**Causal user graphs.** One interpretation of Eq. (1) is that users cope with uncertainty by employing *causal reasoning* [47], this aligning with a predominant approach in the cognitive sciences that views humans as acting based on 'mental causal models' [45]. Here we follow [47] and think of users as reasoning through personalized *user causal graphs*, denoted $G_u$. The structure of $G_u$ expresses $u$'s causal beliefs—namely which variables causally affect others—and its factors correspond to the conditional terms ($p_u$ and $v_u$) in Eq. (1). A key modeling point is that users can vary in their causal perceptions; hence, different users may have different graphs that encode different conditional independencies, these inducing different simplifications of the conditional terms. For example, a user that believes movies with a five-star rating ($r$) are worthwhile regardless of their content ($x$) would have $v_u(x,\bar{x},r)$ reduced to $v_u(\bar{x},r)$, since $v \perp\!\!\!\perp x|r$; meanwhile, a user who, after reading a movie's description ($x$), is unaffected by its rating ($r$), would have $v_u(x,\bar{x})$ instead, since $v \perp\!\!\!\perp r|x$.

### 3.3 User behavior and spurious correlations

We are now ready to make the connection to learning. Recall that our goal is to learn a predictor $f$ that is unaffected by spurious correlations, and that these can materialize if some mechanism creates

an association between $e$ and $y$; we will now see how user behavior can play such a role. Continuing our illustrative example, assume that the beliefs of our boundedly-rational user (who chooses via Eq. (1)) are encoded by the leftmost diagram in Fig. 2 ('User beliefs'). The diagram does not show an edge between $e$ and $y$. However, and crucially, the user behaves 'as if' there actually was an edge: by accounting for uncertainty via integration, $\bar{x}$ is effectively 'removed' from the indirect path $e \rightarrow \bar{x} \rightarrow y$, which results in a direct connection between $e$ and $y$. The corresponding data-generation process of choices $y$ is illustrated in Fig. 2 ('Data generation'). This supports our main argument: by making decisions, users can *generate* spurious correlations in the data—here, by accounting for uncertainty.

The above has concrete implications on learning. First, it shows how conventional learning approaches can fail. On the one hand, since users observe only $x$ and $r$, one reasonable approach would be to discard $\bar{x}$ altogether, and train a predictor $f_u(x, r; \theta)$ in hopes of mimicking user choice behavior. This means learning as if there is no edge between $e$ and $y$ (Fig. 2, 'Model A'). Under this (incorrect) assumption, for $f_u$ to be robust to variation in $e$, it would suffice to train using a conventional approach, e.g., vanilla ERM. But the user *knows* $\bar{x}$ exists, and by integrating beliefs, relies on this for producing $y$—importantly, in a way that *does* depend on $e$. This makes learning $f_u(x, r; \theta)$ prone to using $x$ and $r$ to compensate for the constant effect of each train-time environment $e_k$ on $y$. By definition, once the environment changes, a naïvely trained $f_u$ cannot account for the new (residual) effect of $e$ on $y$.

Conversely, the system may choose to learn using all information that is available to it, namely train a predictor $f_u(x, \bar{x}, r; \theta)$ (Fig. 2, 'Model B'). This make sense if the goal is in-distribution (ID) generalization. But for out-of-distribution, this creates an illusion that conditioning on $x, \bar{x}, r$ will block all paths from $e$ and $y$, again making it tempting to (wrongly) conclude that applying ERM would suffice for robustness. However, because the system does observe instances of $\bar{x}$ (note it still exists in the data generating process), learning can now erroneously use the variation in $\bar{x}$ to explain $y$, whereas the true $y$ does not rely on specific instantiations of $\bar{x}$. As a result, a naïvely learned $f_u$ will likely overfit to training distributions in $\mathcal{E}_{\text{train}}$, and may not generalize well to new environments.

Second, the awareness to how users account for uncertainty suggests a means to combat spuriousness. Eq. (1) shows that $y$ depends on $x, r$, and $e$; hence, since our goal is to discourage the dependence of $\hat{y}$ on $e$, it follows that (i) functionally, $f_u$ should not depend on $\bar{x}$, but (ii) $f_u$ should be learned in a way that controls for (conditional) variation in $e$. In our example, this manifests in the role of $\bar{x}$: at *train-time* use $\bar{x}$ (perhaps indirectly) to learn a function that at *test-time* does not rely on it (c.f. Model B which uses $\bar{x}$ for both train and test, and Model A which does not use $\bar{x}$ at all).[2] Since the precise way in which $e$ relates to other variables is determined by the user graph $G_u$ (which determines conditional independencies), knowledge of the graph should be useful in promoting invariance. In the next section we describe what knowledge is needed, and how it can be used for invariant learning.

## 4 Learning With Causal User Models

Our approach to robust learning is based on regularized risk minimization, where regularization acts to discourage variation in predictions across environments [54,55]. Our learning objective is:

$$\underset{f \in F}{\arg\min} \, L(f; S) + \lambda R(f; S_1, \dots, S_K) \tag{2}$$

where $F$ is the function class, $L$ is the average loss w.r.t an empirical loss function (e.g., log-loss), and $R$ is a data-dependent regularization term with coefficient $\lambda$. In our approach, the role of $R$ is to penalize $f$ for violating certain statistical independencies; the question of *which* independencies should be targeted—and hence the precise form that $R$ should have—can be answered by the underlying causal graph [54]. Knowing the full user graph (e.g. detailed causal relations between different features within $x$, such as whether the genre of a movie is a cause for its production quality) can certainly help , but relying on this (and at scale) is impractical. Luckily, as we show here, coarse information regarding the graph can be translated into necessary conditions for distributional robustness, and in §5 we will see that these can go a long way towards learning robust models in practice.

Note that by choosing to promote robustness through regularization, our approach becomes agnostic to the choice of function class $F$ (although the graph may also be helpful in this[3]). We also need not

---

[2]The careful reader will notice that Fig. 2 reveals how spuriousness can arise, but not yet how it can be handled. Indeed, the latter requires additional structure (which the figure abstracts) that is described in Sec. 4.

[3]E.g., if we aim to learn parameterized predictors on the basis of Eq. (1), graphs can help discard dependencies.

commit to any specific behavioral choice model. This allows us to abstract away from the particular behavioral mechanism that generates spuriousness (e.g., integration of $\bar{x}$), and consider general relations between $e$ and $y$ (e.g., selection or common cause); we make use of this in our experiments.

**Regularization schemes.** We focus on two methods for promoting statistical independence: MMD [22], which we present here; and CORAL [52], which we describe in Appendix C (we use both in our experiments). The MMD regularizer applies to models $f$ that can be expressed as a predictor $h$ applied to a (learned) representation mapping $\phi$, i.e., $f = h \circ \phi$ (note $h$ can be vacuous). MMD works by encouraging the (empirical) distribution of representations for each environment $e_k$ to be indistinguishable from all others; this is one way to express the independence test for $\hat{y}$ and $e$ [54]. In our case, for a single $e_k$, MMD is instantiated as:

$$\text{MMD}(\Phi_k, \Phi_{-k}), \quad \text{where} \quad \Phi_k = \{\phi(x) : x \in S_k\}, \ \Phi_{-k} = \{\phi(x) : x \in S \setminus S_k\}. \tag{3}$$

As we show next, the precise way in which MMD is used for regularization depends on the graph.

## 4.1 User graph classes: causal vs. anti-causal

Following our example in Fig.1a, consider users of two types: a 'causal' user $u_{\to y}$ that believes value is an *effect* of an item's description (i.e., $D^e(x, r, y \mid u = u_{\to y})$ is entailed by the graph $x, r \to y$ for each $e \in \mathcal{E}$), and an 'anti-causal' user $u_{\leftarrow y}$ that believes the item's value *causes* its description (i.e., $D^e(x, r, y \mid u = u_{\leftarrow y})$ is entailed by $x, r \leftarrow y$ respectively).[4] Our next result shows that: (i) $u_{\to y}$ and $u_{\leftarrow y}$ require *different* regularization schemes; but (ii) the appropriate scheme is fully determined by their type—*irrespective* of any other properties of their graphs. Thus, from a learning perspective, it suffices to know which of two classes a user belongs to: causal, or anti-causal.

**Proposition 1.** *Let $f$ be a CI model and assume $y$ and $e$ are confounded (e.g., $e \to y$ exists), then:*

*(1) $f_{u_{\to y}}$ must satisfy $P_{D^e}\left( f_{u_{\to y}}(x, r) \right) = P_{D^{e'}}\left( f_{u_{\to y}}(x, r) \right) \ \forall e, e' \in \mathcal{E}$*

*(2) $f_{u_{\leftarrow y}}$ must satisfy $P_{D^e}\left( f_{u_{\leftarrow y}}(x, r) \mid y \right) = P_{D^{e'}}\left( f_{u_{\leftarrow y}}(x, r) \mid y \right) \ \forall e, e' \in \mathcal{E}, \ y \in \{0, 1\}$*

*On the other hand, $f_{u_{\to y}}$ need not necessarily satisfy (2), and $f_{u_{\leftarrow y}}$ need not necessarily satisfy (1).*

If we fail to enforce these constraints during learning, then we will not learn a CI classifier. On the other hand, enforcing unnecessary constraints (e.g., requiring both conditions hold for $f_{u_{\to y}}$ and $f_{u_{\leftarrow y}}$) restricts our hypothesis class and hence limits performance. The proof follows directly from [54] (under technical assumptions; see Appendix A). The distinction between causal and anti-causal is fundamental in causality [41]; for our purposes, it prescribes the appropriate regularization.

**Corollary 1.** *For any user $u$, to encourage $f_u(x, r)$ to be invariant to changes in $e$, set:*

$$R(f; S) = \begin{cases} \sum_k \text{MMD}(\Phi_{k,u}, \Phi_{-k,u}) & u \text{ is causal} \quad \text{\textit{(marginal MMD)}} \\ \sum_y \sum_k \text{MMD}(\Phi_{k,u}^{(y)}, \Phi_{-k,u}^{(y)}) & u \text{ is anti-causal} \quad \text{\textit{(conditional MMD)}} \end{cases} \tag{4}$$

*where $\Phi_{k,u}, \Phi_k^{(y)}$ includes the subset of examples with user $u$ and label $y$, respectively.*

When learning over multiple users (Eq. (2)), the operational conclusion is that all users of the same class—regardless of their specific graphs—should be regularized in the same manner, as in Eq. (4).

## 4.2 User graph subclasses: inter-feature relations

Consider now two users that are of the same class (i.e., causal or anti-causal), but perceive differently the causal relations between $x$ and $r$: a *believer*, $u_{x \to r}$, who believes recommendations follow from the item's attributes; and a *skeptic*, $u_{x \leftarrow r}$, who presumes that the system reveals item attributes to match a desired recommendation (see Fig.1b). Our main result shows that even if both users share the same objective preferences and hence exhibit the same choice patterns—to be *optimally* invariant, each user may require her own, independently-trained model (though with the same regularization).

**Proposition 2.** *Let $u_{x \to r}, u_{x \leftarrow r}$ be two users of the same class (i.e., causal or anti-causal) but of a different subclass (i.e., believer and skeptic, respectively). Even if there is a single predictor $f$ which is optimal for the pooled distribution $D_{\text{train}}$, each user can have a different optimal CI predictor.*

---

[4]Technically, anti-causal users have $x, r \leftarrow \tilde{v}$, and $y$ as a function of $\tilde{v}$, but we use $x, r \leftarrow y$ for consistency.

Proof is in Appendix A. Prop. 2 can be interpreted as follows: Take some $u$, and 'counterfactually' invert the edges between $x$ and $r$. In some cases, this will have no effect on $u$'s behavior under $\mathcal{E}_{\text{train}}$, and so any $f$ that is optimal in one case will also be optimal in the other. Nonetheless, for optimality to carry over to *other* environments—different predictors may be needed. This is since each causal structure implies a different interventional distribution, and hence a different set of CI predictors: e.g., in $G_{x \leftarrow r}$, the v-structure $e \rightarrow x \leftarrow r$ suggests that an invariant predictor may depend on $r$, yet in $G_{x \rightarrow r}$ it cannot. If the sets of CI predictors do not intersect, then necessarily there is no single optimal model.

The practical take-away is that even if two different users exhibit similar observed behavioral patterns (e.g., differences in their graphs are not expressed in the data), whether they are skeptics or believers has implications for robust learning; Prop. 2 considers an extreme case. Luckily, for data with mixed subclasses, having multiple training environments enables us to nonetheless learn invariant predictors, e.g., by partitioning the data by user subclass—whether inferred or assumed—and learning a different model for each subclass (with regularization determined by class).

**Graph knowledge: inference vs. beliefs.** Formally, both Prop. 1 and Prop. 2 require that we precisely know each user's class and subclass, respectively, which amounts to inferring the directionality of a subset of edges. In principle, this can be done via experimentation (e.g., using focused interventions such as A/B tests) or from observational data (using simple conditional independence tests, e.g., [48,21]) [42]. While this is certainly easier than inferring the entire graph, orienting edges can still be challenging or expensive. Nonetheless, Prop. 1 can still be practical useful when there is good *domain knowledge* regarding user classes, at either the individual or population level: if the learner has certain prior beliefs regarding users' causal perceptions, Eq. (4) provides guidance as to how to devise the learning objective: what regularization to apply (Prop. 1), and how to partition the data (Prop. 2). Our experiments in Sec. 5, which we present next, are designed under this perspective.

# 5 Experiments and Results

We present three experiments: two targeting user classes (*causal* or *anti-causal*) and using real data, and one targeting user subclasses (*believers* and *skeptics*) and using synthetic data. Appendix B includes further details on model architectures, training procedures, and data generation.

## 5.1 Learning with *causal* users: text-based beer recommendation

**Data.** We use *RateBeer*, a dataset of beer reviews with over 3M entries and spanning $\sim$ 10 years [32]. We use the data to generate beer features $x$ (e.g., popularity, average rating) and $r$ (e.g., textual review embeddings) and user features $u$ (e.g., average rating, word counts). Given a sample $(u, x, r)$, our goal is to predict a (binarized) rating $y$. Here we focus on *causal* users, and so would like labels $y$ to expresses causal user beliefs. The challenge is that our observational data is not necessarily such. To simulate causal user behavior, we rely on the observation that $x, r \rightarrow y$ means "changes in $x, r$ affect $y$", and for each $u$ create an individualized empirical distribution of 'counterfactual' samples $(x', r', y')$ that approximate the entire intervention space (i.e., all counterfactual outcomes $y'$ under possible interventions $(x, r) \mapsto (x', r')$). Training data is then generated by sampling from this space.

We consider each year as an environment $e$, with each $e$ inducing a distribution over $(u, x, r)$. We implement spuriousness via selection: Each $e$ entails different fashionable 'tastes' in beer, expressed as a different weighting over the possible beer types (e.g., lager, ale, porter). Labels are then made to correlate with tastes in a certain temporal pattern. This serves as a mechanism for spurious correlation.

**Results.** Fig. 3 compares the performance over time of three training procedures that differ only in the type of regularization applied: *causal*, *anti-causal*, and *non-causal*. Our data includes behavior generated by causal-class users; results demonstrate the clear benefit of using a behaviorally-consistent regularization scheme (here, causal). Note the causal approach is not optimal in 2006 and 2008; this is since correlations in $e \leftrightarrow y$ are set to make these years similar to the training data. However, in the face of significant shifts in taste, other approaches collapse, while the causal approach remains stable.

## 5.2 Learning with *anti-causal* users: clothing-style recommendation

**Data.** We use the *fashion product images* dataset[5], which includes includes 44.4$k$ fashion items described by images, attributes, and text. Here we focus on *anti-causal* users, and generate data

---

[5]https://www.kaggle.com/paramaggarwal/fashion-product-images-dataset

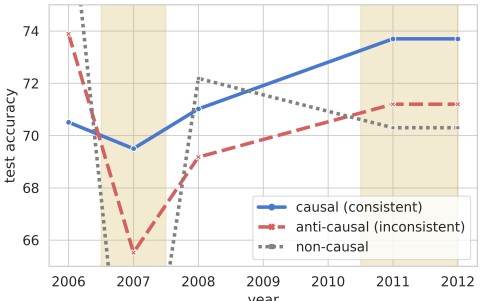

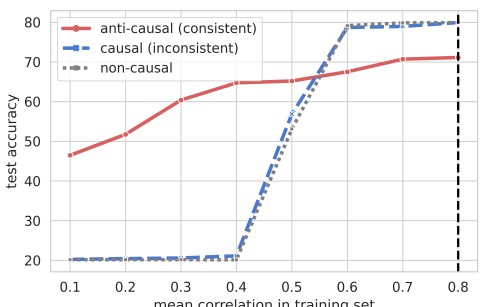

Figure 3: ***RecBeer* Results**. For each year, models are trained on past data (starting 2002), and predict on the following year. The *causal* training scheme, consistent with the user class, outperforms other methods when beer-type fashions ($e$) changes. Periods with substantial change are highlighted in tan.

Figure 4: ***RecFashion* Results**. Environments vary in the correlation between item colors and user choices. The *anti-causal* regularization scheme, consistent with the user class, outperforms methods when test-time deviates from train-time correlation ($=0.8$). When correlations flip ($< 0.5$), other methods crash.

in a way similar to §5.1, but using an anti-causal intervention space. In this experiment we let user choices $y \in \{0, 1\}$ depend on an item's image and color, which can be either red or green; in this way, $x$ is the item's grayscale image, and $r$ its hue (which we control). Here we consider environments $e$ that induce varying degrees of spurious correlations between color and user choices, $P(y = 1|\text{red}) = P(y = 0|\text{green}) = p_e$. For the test set we use $p_e = 0.8$, and experiment with training data that gradually deviate from this relation, i.e., having $p_{e'} \in [0.1, 0.8]$.

**Results.** Fig. 4 shows that consistent regularization (here, anti-causal) outperforms other alternatives whenever correlations deviate from those observed in training. Once correlations flip ($< 0.5$), both *causal* and *non-causal* approaches fail catastrophically; the *anti-causal* approach remains robust.

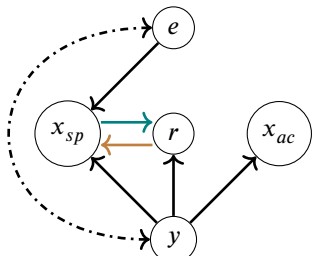

Figure 5: Data-generating process for the user subclass experiment (synthetic). Here, $x$ factorizes into an *anti-causal* component $x_{ac}$, and a spurious component $x_{sp}$ linked with $r$. Spuriousness results from selection bias between $y$ and $e$.

Table 1: Accuracy for the user subclass experiment. Rows show train conditions: with and without regularization, and which users are included in the training set. Columns show test conditions: ID/OOD, and user type. Best results for each train condition (rows) are highlighted in bold.

| Reg. | Users@train | Accuracy (ID / OOD) | |
|---|---|---|---|
| | | *skeptic* | *believer* |
| $\lambda = 0$ | *skeptic* | **78.0** / 50.0 | **89.8** / 75.1 |
| | *believer* | **78.0** / 50.0 | **89.8** / 75.1 |
| | both | **78.0** / 50.0 | **89.8** / 75.1 |
| $\lambda > 0$ | *skeptic* | 71.1 / **75.6** | 74.67 / 75.5 |
| | *believer* | 69.8 / 52.5 | 88.5 / **85.2** |
| | both | 70.03 / 64.57 | 78.88 / 78.13 |

### 5.3 Learning with multiple user subclasses

Our final experiment studies learning with users of of the same-class (here, *anti-causal*) but different subclasses: *skeptics* or *believers*. Our analysis in §4.2 suggests that each user subclass may have a different optimal predictor; here we investigate this empirically on synthetic data.

**Data.** The data-generating process is as follows (see Fig. 5.2). We use three environments: $e_1, e_2$ at train, and $e_3$ at test, and implement a selection mechanism (dashed line) that causes differences in $p(y|e_i)$ across $e_i$. Since we focus on anti-causal users, features $x, r$ are determined by $e, y$. We use three binary features: $x_{sp}$ ('spurious'), $x_{ac}$ ('anti-causal'), and $r$. These are designed so that an $f$ which uses $x_{ac}$ alone obtains 0.75 accuracy, but using also $x_{sp}$ improves *in-distribution* (ID) accuracy slightly to 0.78, and so the optimal ID predictor for both user subclasses is of the form $f^*(x_{ac}, x_{sp})$. However, relying on $x_{sp}$ causes *out-of-distribution* (OOD) performance to deteriorate considerably;

thus, robust models should not learn to discard $x_{sp}$. The role of $r$ is to distinguish between user subclasses: The skeptic does not need $r$ since, for her, it is fully determined by $x_{sp}$; the optimal invariant predictor is hence $f_{x \to r}(x_{ac})$. Meanwhile, the believer, due to the v-structure $r \to x_{sp} \leftarrow e$, can benefit in-distribution by using both $r$ and $x_{sp}$; here, the optimal invariant predictor is $f_{x \leftarrow r}(x_{ac}, r)$.

**Results.** Table 1 shows ID and OOD performance for each user subclass (columns), for learning with and without regularization (rows). Since all users are anti-causal, we use anti-causal (i.e., conditional) regularization. We compare learning a separate predictor for each user type (rows '*skeptic*' and '*believer*') and learning a single predictor over all users jointly ('both'). Results show that without regularization ($\lambda = 0$), ID performance is good, but the learned predictor fails OOD—drastically for skeptic users (note all rows are the same since both user types share the same ID-optimal $f^*$). In contrast, when regularization is applied ($\lambda > 0$), learning an independent predictor for each user subclass performs well OOD (for both subclasses), indicating robustness to changing environments; note that ID performance is also mostly maintained. Meanwhile, learning on the entire dataset (i.e., including both user types) does provide some robustness—but is suboptimal both ID and OOD.

## 5.4 Learning with mixed sub-populations

Our previous experiment considered a setting in which the learner has exact information regarding each user's sub-type, and so can correctly partition the population in a way that is optimal in regards to Prop. 2. However, such precise knowledge may not be available in practice, or may be too costly or difficult to infer. In this section we experiment in a setting where the learner has only coarse information (or general beliefs) about user (sub-)types. Our results suggests that following the practical conclusions of Prop. 2—namely partitioning the population of users based on (estimated) types and learning a different predictive model for each—can be beneficial even when based only on a reasonable guess.

**Data.** We use the setting of Sec. 5.3 with a population of anti-causal users, composed of two subpopulations of skeptics and believers. We then simulate a setting where the learner has imprecise information about user types by adding noise: we move an $\alpha$-fraction of each subpopulation into the other, this creating two mixed sub-populations. with increasing levels of 'impurity', $\alpha \in [0, 0.25]$. Here, $\alpha = 0$ represents perfect information, whereas $\alpha = 0.25$ entails large minority groups (25%).

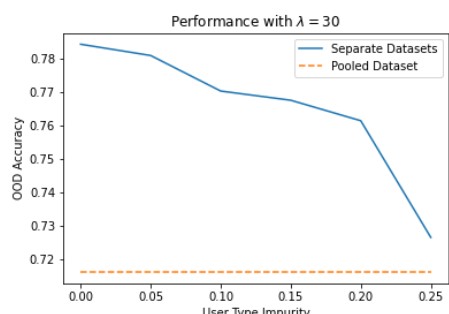

**Results.** Fig. 6 compares OOD performance of the robust model (solid line) to a naïve model trained on the pooled dataset (dashed line). In line with previous results, for $\alpha = 0$, the robust model achieves significantly higher accuracy on the test environment. As $\alpha$ increases, the robust model preserves its advantage, with accuracy degrading gracefully; for $\alpha = 0.25$, the robust model still outperforms the pooled baseline. Thus, our results show that despite having imperfect information regarding user sub-types, learning distinct models for each subpopulation, as Prop. 2 suggests, remains beneficial.

Figure 6: Learning with a mixed population of believers and skeptics. Even when the minority group is large (25%), learning in a way that is tailored to the majority group is still beneficial.

## 6 Discussion

Humans beings perceive the world causally; our paper argues that to cope with a world that *changes*, learning must take into account how humans believe these changes take effect. We identify one key reason: in making decisions under uncertainty, users can *cause* spurious correlations to appear in the data. Towards this, we propose to employ tools from invariant causal learning, but in a way that is tailored to how humans make decisions, this drawing on economic models of bounded-rationality. Our approach relies on regularization for achieving invariance, with our main point being that *how* and *what* to regularize can be derived from users' causal graphs. Although we have argued that even partial graph information can be helpful—even this form of knowledge is not straightforward to obtain (notably at test-time), and may require experimentation. Nonetheless, and in hopes of spurring further interest, we view our work as taking one step towards establishing a disciplined causal perspective on the interaction between recommending systems and the decision-making users they aim to serve.

## Acknowledgements

This work was supported in part by The Israel Science Foundation (grant 278/22).

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
