# A  Details on Formal Claims

Our claim in Proposition 1 is also based on the setting of Veitch et al. [54]. Under the assumption that $e$ is discrete, Lemma 3.1 of [54] ensures that there exists a random variable $(x,r)_e^\perp$ such that $f_u(x,r)$ is CI if and only if it is $(x,r)_e^\perp$-measurable. Then we will assume that $x,r$ can be decomposed into parts $x, r_{y \wedge e}, x, r_y^\perp, x, r_e^\perp$. Note that we do not assume that we know how to decompose our features in this manner, nor we assume anything about the semantic meaning of these components. We only assume that this decomposition exists, and then the main assumption made in [54] is that the graph in Fig. 1a conforms to the structures in Figure 7 for each user type.

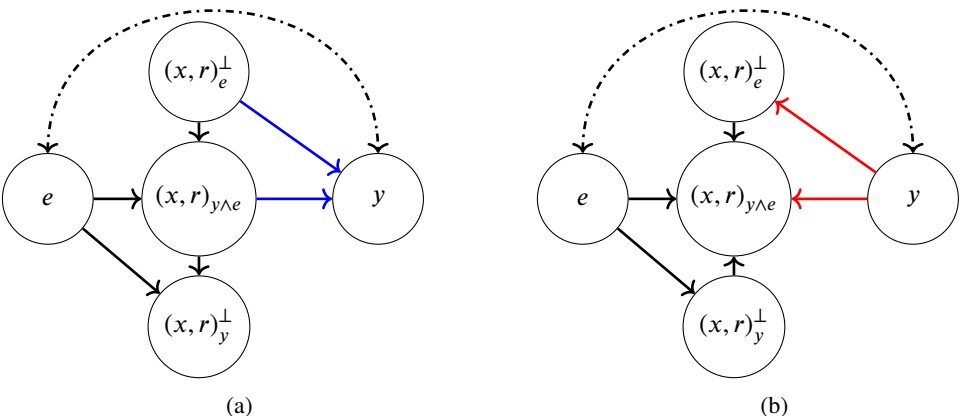

$$(a) \qquad\qquad\qquad (b)$$

Figure 7: Detailed graphs describing our assumptions on causal and anti-causal users (a) causal model for data generating process of *causal* user, and (b) *anti-causal* user. Dashed lines indicate possible confounding.

We are now ready to state Proposition 1 in a more precise manner

**Proposition 3.** *Let $f$ be a CI model and assume $y$ and $e$ are confounded (i.e. they are connected by an unobserved common cause $c$ or by a directed path). Further assume that $D^e(x,r,y \mid u)$ is entailed by the causal models in Fig. 7 for $u = u_{\to y}$ and $u = u_{\leftarrow y}$. Then the following holds:*

1. *$f_{u_{\to y}}$ must satisfy $D^e(f_{u_{\to y}}(x,r)) = D^{e'}(f_{u_{\to y}}(x,r)) \ \ \forall e, e' \in \mathcal{E}$.*
2. *$f_{u_{\leftarrow y}}$ must satisfy $D^e(f_{u_{\leftarrow y}}(x,r) \mid y) = D^{e'}(f_{u_{\leftarrow y}}(x,r) \mid y) \ \ \forall e, e' \in \mathcal{E}, \ y \in \{0,1\}$.*

*On the other hand, $f_{u_{\leftarrow y}}$ and $f_{u_{\to y}}$ do not necessarily satisfy conditions 1 and 2, respectively.*

*Proof.* Under the assumptions laid out about the causal model, the conditional independence relations can be read off the graph directly, as in Theorem 3.2 of [54]. This proves that the independence properties stated in the proposition must hold. To see that $f_{u_{\leftarrow y}}, f_{u_{\to y}}$ do not necessarily satisfy properties 1 and 2 respectively, we will prove the existence of such cases. Consider a causal model where $e$ and $y$ are confounded, and assume that the model is faithful [35] (i.e. all conditional independence statements that are not entailed by the graph do not hold). Hence for the causal user we generally have $D^e((x,r)_e^\perp \mid y, u = u_{\to y}) \neq D^{e'}((x,r)_e^\perp \mid y, u = u_{\to y})$ (at the very least there are values of $(x,r)_e^\perp, y$ for which this holds), and hence there exists some $(x,r)_e^\perp$-measurable function $\hat{f}_{u_{\to y}}(x,r)$ that satisfies $D^e(f(\hat{x},r) \mid y, u = u_{\to y}) \neq D^{e'}(f(\hat{x},r) \mid y, u = u_{\to y})$. The same argument can be applied for the anti-causal user $u_{\leftarrow y}$ to prove the existence of an $(x,r)_e^\perp$-measurable function $\hat{f}_{u_{\leftarrow y}}(x,r)$ that satisfies $D^e(f(\hat{x},r) \mid u = u_{\leftarrow y}) \neq D^{e'}(f(\hat{x},r) \mid u = u_{\leftarrow y})$. The model $\hat{f}(x,r)$ is CI since the constructed functions are $(x,r)_e^\perp$-measurable, but models $f_{u_{\leftarrow y}}, f_{u_{\to y}}$ do not satisfy conditions 1 and 2 respectively, which concludes our claim. $\square$

Next we prove Proposition 2 by constructing a confounded model for an anti-causal user, similar to the one in the synthetic experiment of Section 5.3. Towards this proposition, we point out that

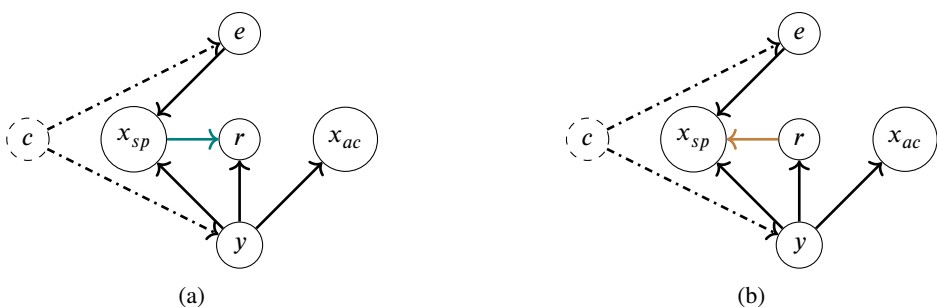

Figure 8: Graphs describing the data-generating processes for anti-causal *believer* and *skeptic* users in the proof of Proposition 2.

an optimal CI predictor is defined as a CI predictor with the best possible worst case performance. Where the worst case is taken over all distributions that are causally-compatible [54] with the source distribution $D_{\text{train}}$.

**Definition 2.** *$D_{train}$ and $D_{OOD}$ are causally compatible if they are entailed by the same causal graph, $D_{train}(y) = D_{OOD}(y)$, and there is a confounder $c$ and/or selection conditions $s, \tilde{s}$ such that $D_{train} = \int D_{train}(x_{sp}, x_{ac}, r, y \mid c, s = 1)d\tilde{P}(c)$ and $D_{OOD} = \int D_{train}(x_{sp}, x_{ac}, r, y \mid c, \tilde{s} = 1)d\tilde{Q}(c)$ for some $\tilde{P}(c), \tilde{Q}(c)$.*

Let us focus now on distributions where $f(x_{sp}, r, x_{ac})$ is counterfactually invariant if and only if it is $(r, x_{ac})$-measurable (the expression $(r, x_{ac})$ should be read as a bivariate random variable). Note again that from Lemma 3.1 of [54] such a variable exists. The following claim will help us reason about the optimal CI model for users of the skeptic sub-class.

**Lemma 1.** *If $D_{train}$ is entailed by the graph in Fig. 8a and $D_{OOD}$ is causally compatible with it, then $D_{train}(y \mid r, x_{ac}) = D_{OOD}(y \mid r, x_{ac})$.*

*Proof.* For binary classification, it is enough to show that $\frac{D_{\text{train}}(y=1|r,x_{ac})}{D_{\text{train}}(y=0|r,x_{ac})} = \frac{D_{\text{OOD}}(y=1|r,x_{ac})}{D_{\text{OOD}}(y=0|r,x_{ac})}$. Let us write this for the training distribution:

$$\frac{D_{\text{train}}(y = 1 \mid r, x_{ac})}{D_{\text{train}}(y = 0 \mid r, x_{ac})} = \frac{D_{\text{train}}(r, x_{ac} \mid y = 1)D_{\text{train}}(y = 1)}{D_{\text{train}}(r, x_{ac} \mid y = 0)D_{\text{train}}(y = 0)}$$
$$= \frac{D_{\text{train}}(r, x_{ac} \mid y = 1)D_{\text{OOD}}(y = 1)}{D_{\text{train}}(r, x_{ac} \mid y = 0)D_{\text{OOD}}(y = 0)}.$$

The second equality stems from the causal-compatibility of $D_{\text{OOD}}$. It is left to show that $D_{\text{train}}(y \mid r, x_{ac}) = D_{\text{OOD}}(y \mid r, x_{ac})$. From causal-compatibility the distributions are entailed by the same graph in Fig. 8a, which imposes the conditional independence $c \perp r, x_{ac} \mid y$. Hence we conclude the proof by:

$$D_{\text{train}}(x_{ac}, r \mid y) = \int D_{\text{train}}(x_{ac}, r \mid y, c)d\tilde{P}(c) = \int D_{\text{train}}(x_{ac}, r \mid y, c)d\tilde{Q}(c) = D_{\text{OOD}}(x_{ac}, r \mid y).$$

$\square$

From this result we gather that if we only consider the features $x_{ac}, r$, there is a unique Bayes-optimal classifier over all target distributions that are causally compatible with $D_{\text{train}}$. Since a classifier is CI if and only if it is $(x_{ac}, r)$-measurable, we see that for the skeptic sub-class of users the optimal CI model is $f(x_{sp}, r, x_{ac}) = D_{\text{train}}(y \mid r, x_{ac})$. The rest of the proof will simply show that this model may not be CI for a user of sub-type *believer* that has the same choice patterns over observed data pooled from two training environments.

*Proof of Proposition 2.* Consider a data generating process as depicted in Figure 8a. All variables $x_{sp}, r, x_{ac}, y, c$ are binary, we consider 2 training environments $\mathcal{E}_{\text{train}} = \{0, 1\}$. We write down the

distribution in a factorized form:

$$D_{u_{x\leftarrow r}}(x_{sp}, x_{ac}, r, y) = \sum_{c\in\{0,1\}, e\in\{0,1\}} p(c)p(y\mid c)p(x_{ac}\mid y)p(e\mid c)p_{u_{x\leftarrow r}}(r\mid y)p^e_{u_{x\leftarrow r}}(x_{sp}\mid r, y)$$

$$= p(x_{ac}\mid y)p_{u_{x\leftarrow r}}(r\mid y)\left(\sum_{e\in\{0,1\}}\tilde{p}(e, y)p^e_{u_{x\leftarrow r}}(x_{sp}\mid r, y)\right).$$

Here we defined $\tilde{p}(e, y) = \sum_{c\in 0,1} p(y, c)p(e\mid c)$. The subscripts $u_{x\leftarrow r}$ emphasize that in the distribution we will construct for the believer user, $D_{u_{x\to r}}$, all factors that are not subscripted will be equal to those in $D_{u_{x\leftarrow r}}$. That is, consider a distribution that factorizes over the graph in Figure 8b as follows:

$$D_{u_{x\to r}}(x_{sp}, x_{ac}, r, y) = p(x_{ac}\mid y)p_{u_{x\to r}}(r\mid y, x_{sp})\left(\sum_{e\in\{0,1\}}\tilde{p}(e, y)p^e_{u_{x\to r}}(x_{sp}\mid y)\right). \tag{5}$$

We will show that there exists some setting of $p_{u_{x\to r}}(r\mid y, x_{ac})$, $p^e_{u_{x\to r}}(x_{sp}\mid y)$ such that:

$$D_{u_{x\leftarrow r}}(x_{sp}, x_{ac}, r, y) = D_{u_{x\to r}}(x_{sp}, x_{ac}, r, y).$$

But it will also satisfy $D^0_{u_{x\to r}}(y\mid r, x_{ac}) \neq D^1_{u_{x\to r}}(y\mid r, x_{ac})$. Then the proof will be concluded, as $f(x_{sp}, x_{ac}, r) = D_{u_{x\leftarrow r}}(y\mid r, x_{ac}) = D_{u_{x\to r}}(y\mid r, x_{ac})$ cannot be CI w.r.t $D_{u_{x\to r}}$. This holds since $D^e_{u_{x\to r}}(y\mid r, x_{ac}) \neq D_{u_{x\to r}}(y\mid r, x_{ac})$ for $e\in\{0,1\}$, hence there must be some instance for which $f(x_{ac}(0), x_{sp}(0), r(0)) \neq f(x_{ac}(1), x_{sp}(1), r(1))$.

Towards this, consider $D_{u_{x\leftarrow r}}(r\mid y, x_{sp})$ which is obtained by the respective marginalization and conditioning of $D_{u_{x\leftarrow r}}(x_{sp}, x_{ac}, r, y)$, and also consider $\sum_{e\in 0,1}\tilde{p}(e, y)D^e_{u_{x\leftarrow r}}(x_{sp}\mid y)$. Let us set:

$$p_{u_{x\to r}}(r\mid y, x_{sp}) := D_{u_{x\leftarrow r}}(r\mid y, x_{sp}).$$

It is clear that if we set $p^e_{u_{x\to r}}(x_{sp}\mid y)$ such that the following holds:

$$\sum_{e\in\{0,1\}}\tilde{p}(e, y)p^e_{u_{x\to r}}(x_{sp}\mid y) = \sum_{e\in\{0,1\}}\tilde{p}(e, y)D^e_{u_{x\leftarrow r}}(x_{sp}\mid y), \tag{6}$$

then the equality $D_{u_{x\leftarrow r}}(x_{sp}, x_{ac}, r, y) = D_{u_{x\to r}}(x_{sp}, x_{ac}, r, y)$ also holds. That is because the factorization in (5) is a factorization of the joint distribution over $x_{sp}, x_{ac}, r, y$ where all factors are equal to the ones obtained from $D_{u_{x\leftarrow r}}(x_{sp}, x_{ac}, r, y)$. [6]

Finally, we claim that many solutions satisfy (6). For each value of $y, x_{sp}$ Eq. (6) is a linear equation with two variables ($p^0_{u_{x\to r}}(x_{sp}\mid y)$ and $p^1_{u_{x\to r}}(x_{sp}\mid y)$), and they should be constrained to take values in the range [0, 1]. One solution to the equation is to set $p^e_{u_{x\to r}}(x_{sp}\mid y) := D^e_{u_{x\leftarrow r}}(x_{sp}\mid y)$, and unless $D^e_{u_{x\leftarrow r}}(x_{sp}\mid y)\in\{0,1\}$ for each value of $x_{sp}, y$, and $D^0_{u_{x\leftarrow r}}(x_{sp}\mid y) = D^1_{u_{x\leftarrow r}}(x_{sp}\mid y)$ (i.e. the spurious feature completely determines $y$) the set of solutions to the equations forms an interval in $\mathbb{R}^2$, and has Lebesgue measure that is non-zero.

Thus let us consider the set of parameterized (by the factors in (5)) distributions $\tilde{D}_{u_{x\to r}}(e, x_{sp}, r, x_{ac}, y)$ that satisfy $\sum_{\tilde{e}}\tilde{D}_{u_{x\to r}}(e = \tilde{e}, x_{sp}, r, x_{ac}, y) = D_{u_{x\leftarrow r}}(x_{sp}, r, x_{ac}, y)$ for the fixed distribution $D_{u_{x\leftarrow r}}(x_{sp}, r, x_{ac}, y)$. This set has a non-zero Lebesgue measure over the linearly independent parameters needed to parameterize $D_{u_{x\leftarrow r}}$. Since the set of parameters that yield unfaithful distributions w.r.t a graph has Lebesgue measure zero [49], there must be at least one distribution $\tilde{D}_{u_{x\to r}}(e, x_{sp}, r, x_{ac}, y)$ in the set where the independence $r, x_{ac}\perp e\mid y$ does *not* hold. For such a distribution we will have $D^e_{u_{x\to r}}(y\mid r, x_{ac}) \neq D_{u_{x\to r}}(y\mid r, x_{ac})$, which is what was required to conclude the proof.

$\square$

# B   Experimental Details

Code and data for all experiments can be found in the following anonymous link:
https://drive.google.com/drive/folders/1bO57v4PUuUh76F_q0a_xAVx6CKdeDJ5l

---

[6]Note that it is easy to observe that the two sides of (6) are the marginal distribution over $x_{sp}, y$ of the two distributions $D_{u_{x\to r}}$ and $D_{u_{x\leftarrow r}}$ respectively.

| Table 2: Original *RateBeer* dataset statistics. | |
| --- | --- |
| Number of reviews | 2, 924, 127 |
| Number of users | 40, 213 |
| Number of beers | 110, 419 |
| Users with > 50 reviews | 4, 798 |
| Median #words per review | 54 |
| Timespan | 4/2000-11/2011 |

Table 3: Our *RecBeer* data features.

| Variable type | Not. | Description |
| --- | --- | --- |
| Item | $x$ | avg past appearance
avg past aroma
avg past palate
avg past taste
# of active years
alcohol percentage
beer type |
| User | $u$ | avg past satisfaction
# of past choices
# of active years |
| Recommendation | $r$ | text review
# of past reviews |
| Time | $e$ | year |
| Choice | $y$ | try beer/not |

## B.1 *RecBeer* (causal users)

**Original Dataset description.** The original *RateBeer* dataset includes textual reviews and numerical ratings of roughly 3000 unique beers, collected over the span of over 11 years. Each review data-point also includes additional features describing the beer (e.g., brand, style), the author of the review (e.g., location), and the review itself (e.g., date). Figure 9 shows an example of a data point. Table 2 provides summary statistics.

**Data Generation Process.** The original RateBeer dataset includes reviews and rating that were authored and submitted by users of the platform. For our purposes, focusing learning and prediction on users as *contributors* of content has two limitations: (i) we cannot know what platform-selected information ($r$) was presented to them and how it influenced their decisions, and (ii) we cannot reason counterfactually about their potential choices had they been exposed to different information.

To overcome both issues, we adapt the original dataset to simulate choice behavior of users as *consumers* of content, as they use the platform to make informed decisions about beer consumption. We emulate the following process: a user $u$ logs on to the platforms, and is recommended a certain beer. The beer is described by intrinsic features $x$, and one platform-selected textual review $r$, chosen from a pool of already-existing reviews for that beer (these being the reviews for that beer that have already by submitted by other contributing users). The user then decides weather to try (i.e., consume) the beer ($y = 1$) or not ($y = 0$). Our goal is to predict for new users $u$ their choices $y$ for recommended beers given descriptions $x, r$.

To create features for beers $x$ and (consuming) users $u$, we aggregate information from all corresponding reviews: for beers—all reviews of that beer, and for users—all reviews authored by that user. This includes features such as average past taste score for beers and average past overall satisfaction for users. Table 3 summarizes our feature space. Since we model users as *causal*, the graph edge $r \rightarrow y$ implies that changes to $r$ causally affect $y$. To simulate this behavior, we create for each user an 'intervention space' which includes a collection of possible interventions $r$ and their corresponding counterfactual outcomes $y$. For our experiment, we simply take all pairs of reviews and ratings $(r, s)$ for a given beer to be the set of possible interventions and outcomes. Textual reviews are featurized using a pre-trained BERT model [13], and numerical ratings $s \in [0, 5]$ are transformed into binary choices $y = \{0, 1\}$ by setting $y = 1$ if the user's rating for that beer was above the median rating (for that beer), and $y = 0$ otherwise. Since learning requires observational data, for each user-beer pair $(u, x)$ we sample (in a way we describe shortly) one review-choice pair $(r, y)$ out of 100 unique reviews for that beer; an example is presented in Figure 10. This provides a sampled tuple $(u, x, r, y)$ expressing the behavior of a *causal* user whose choices are affected by the review presented to her. Together, $u, x$, and $r$ (as an embedding) include 866 features.

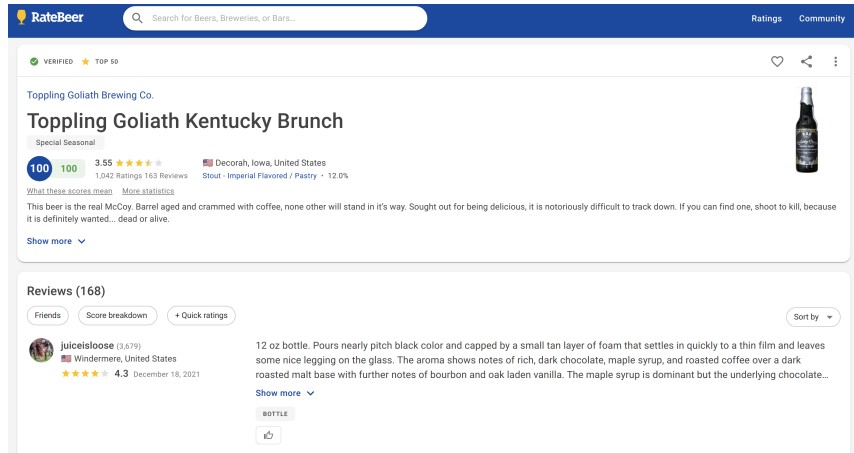

Figure 9: *RateBeer* **example:** A textual review and numerical rating for a beer (with metadata).

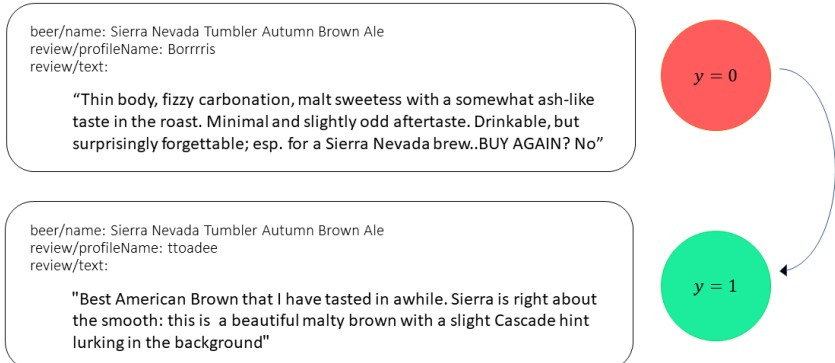

Figure 10: *RecBeer* **interventions:** An example of a simulated intervention for causal users, for which changing the review shown to the user (bottom) to another (top) may influence his behavior (here, from not choosing to choosing).

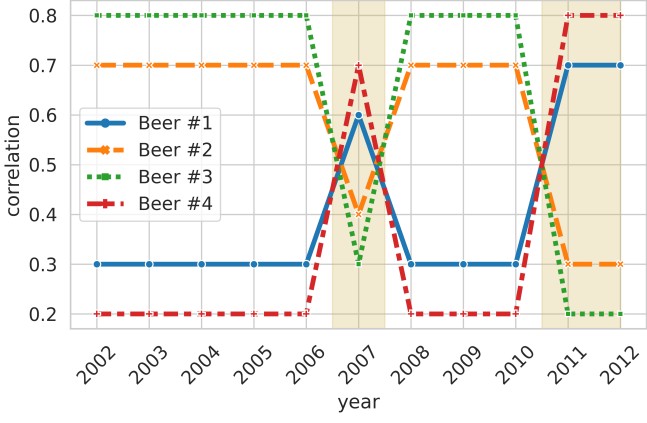

Figure 11: *RecBeer* **environments:** Each year serves as a different environment, whose affect is expressed through differing correlations between beer types and user choices. The plot shows the temporal correlation structure used for the experiment in §5.1, and underlie the results presented in Fig. 3. Periods with substantial changes are highlighted in tan.

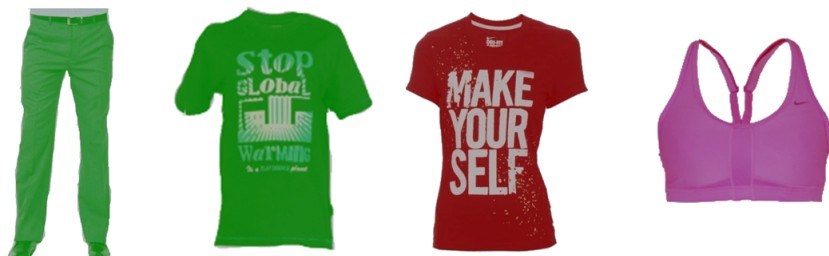

Figure 12: Fashion items in the *RecFashion* dataset with recommended colors. On the left side are green recommendations and on the right side are red recommendations.

Finally, to model the effects of changing environments, we consider an environment variable $e$ that encodes the year, expressing the idea that different years may express different 'trends' in which beer *types*[7] are more (and less) fashionable. To implement this, we sample review-choice pairs for users within each year in a way that introduces a pre-determined amount of correlation between choices and beer types. The chosen per-year correlation levels is plotted in Figure 11. Notice the drastic change in fashions in 2007 and 2011.

**Training and testing.** We train and evaluate one model per year. For each year $e \in \{2006, \dots, 2012\}$, training is performed on data from years $\{2002, \dots, e-1\}$ and tested on $e$. In this way, fashions regarding beer type accumulate over time.

**Models.** We learn a linear model that takes as input the concatenation of $u, x, r$. The learning objective includes a binary cross entropy loss, and marginal MMD as regularization [22] (since we model users as causal; see §4). We trained all models for 700 epochs with $lr = 0.01$ and batches of size 1024, and set $\lambda = 100$. Results are averaged over five runs with different random seeds.

### B.2 *RecFashion* (anti-causal users)

**Original Dataset Statistics.** The *Fashion Product Images* dataset includes a large collection of fashion items, described by an image and additional attributes such as: season, gender, base color, usage, year, and product display name. Items are organized by category, sub-category, and type; we focus on the *apparel* category. Table 4 provides summary statistics.

Table 4: Original *Fashion Product Images* dataset statistics.

| | |
|---|---|
| number of items | $44,447$ |
| main categories | 7 |
| sub-categories | 45 |
| types | 142 |

**Data Generation Process.** The original dataset does not include user choices (or any other form of user behavior). To simulate user choices, we imagine a setting were the platform recommends to each user an item by presenting an image of the item ($x$) in a certain color ($r$). We set $x$ to be the item's grayscale image, and set $r$ to be a colorization of that image into one of two colors: red or green. Users then choose whether to buy the item or not, $y \in \{0, 1\}$. We then model users as choosing primarily on the basis of the 'gender' attribute of items, $x_g \in \{0, 1\}$, and set $y = x_g$ w.p. 0.75 and $y = 1 - x_g$ otherwise.

Since users in this experiments are anti-causal, they act under the belief that changes in $y$ affect $r$ (here we do not make use of the edge $y{\rightarrow}x$). Note that $e$ also affects $r$. We implement this joint influence of $e, y$ on $r$ by assigning colors to images in a way that obtains a certain level of correlation between the color $r \in \{\text{red}, \text{green}\}$ and choices $y$. Technically, we associate with each environment

---

[7]We create four beer 'types' by aggregating beers of similar style. For example, the styles *Doppelbock*, *Dortmunder*, *Dunkel*, *Dunkelweizen*, and *Dunkler* were all attributed to the same type.

$e$ a parameter $p_e \in [0, 1]$. Then, using a color variable $c = 0$ for red and $c = 1$ for green we assign for each item its color as $c = y$ w.p. $p_e$, and $c = 1 - y$ otherwise. Thus, different environments entail different conditional distributions $P(r = \text{red}|y = 1) = P(r = \text{green}|y = 0) = p$, which reflect an anti-causal structure. Finally, given the sampled $c$, we colorize the image $x$ as follows: if $c = 1$, we set $x_R \leftarrow 0.5 + 0.2x_R$, $x_G \leftarrow 0.7x_G$, $x_B \leftarrow 0.7x_B$; if $c = 0$, we set $x_G \leftarrow 0.5 + 0.2x_G$, $x_R \leftarrow 0.7x_R$, $x_B \leftarrow 0.7x_B$ ($R, G, B$ are the color channels). Note that this means users do not observe $x, r$ independently, but rather a colored image that is a product of both $x$ and $r$.

**Training and testing.** We run eight experiments that differ in the average degree of correlation in the training sets, for average correlation values of $p \in \{0.1, 0.2, \ldots, 0.8\}$. Each experimental condition ($p$) includes training data from six environments $e$, with correlations $p_e n v \in \{p - 0.025, p + 0.025, p - 0.05, p + 0.05, p - 0.1, p + 0.1\}$ (their average is $p$).

**Models.** For the model We used a feed forward neural network with three hidden layers and a hidden dimension of size 256, ReLU activation function and $NLL$ as our base loss function. For computational efficiency, input images were resized to $14 \times 14$. The learning objective includes a binary cross entropy loss, and a conditional DeepCORAL regularizer [52] (since we model users as anti-causal; see §4). We set $\lambda = 5000$ in the first 125 epochs and $\lambda = 1$ in the rest, and trained the model for 1,900 epochs with $lr = 0.001$ and batches of size 1024.

# C Loss Functions.

We train all of our models with either the *CORAL* or *MMD* loss. Empirically, we found that *CORAL* we more stable in the *RecFashion* experiments and. In the *RecBeer* experiments, models trained with the *MMD* loss consistently outperformed those who were not. When conditioning on the label $y$, we compute $l_{dist}$ (either $l_{CORAL}$ or $l_{MMD}$) separately for cases where $y = 1$ and $y = 0$. We describe here both loss functions.

*CORAL* **Loss.** The *CORAL* loss is the distance between the second-order statistics of two feature representations, corresponding to different $z$:

$$l_{CORAL}(f(x, r), z) = \frac{1}{d^2}||C_z - C_{z'}||_F^2$$

where $|| \cdot ||_F^2$ denotes the squared matrix Frobenius norm. The covariance matrices of the source and target data are given by:

$$C_z = \frac{1}{n_z - 1}(\phi(x(z), r)^\top \phi(X(z), r)$$
$$- \frac{1}{n_z}(\mathbf{1}^\top \phi(x(z), r))^\top (\mathbf{1}^\top \phi(x(z), r)))$$

where $\mathbf{1}$ is a column vector with all elements equal to 1, and $\phi(\cdot)$ is the feature representation.

*MMD.* Maximum mean discrepancy (*MMD*) measures distances between mean embeddings of features. That is, when we have distributions $P$ and $Q$ over a set $\mathcal{X}$. The *MMD* is defined by a feature map $\phi : \mathcal{X} \to \mathcal{H}$, where $\mathcal{H}$ is what's called a reproducing kernel Hilbert space. In general, the *MMD* is

$$\text{MMD}(P, Q) = ||\mathbb{E}_X[\phi(X)] - \mathbb{E}_Y[\phi(Y)]||_\mathcal{H}$$

For use of the MMD loss for causal representation learning, see Veitch et al. [54].