# OpenReview forum: "In the Eye of the Beholder: Robust Prediction with Causal User Modeling"
_NeurIPS.cc/2022/Conference — NeurIPS 2022 Accept_

### Official Review · Reviewer_HSTu · 2022-07-10

**Rating:** 5
**Confidence:** 4
**Soundness:** 3 good
**Presentation:** 3 good
**Contribution:** 2 fair

**Summary:**

The paper considers a recommendation setting where we observe the attributes of users, of items, and aspects of the recommendation itself such as a relevance, an explanation of the relevance, etc. The task is to use this information to predict the user's decision, whether or not they will interact with the item of interest. The decision could be binary, a rating, or something else.

The problem is to produce a predictor of user decisions that is invariant to changes in the distribution of the observed features (attributes, recommendation aspects, etc.). We expect such invariant predictors to achieve bounded risk when deployed to unseen distributions of data. This paper closely follows work on counterfactual (CF) invariance, a criteria based on counterfactual distributions that captures desirable behavior in predictors. The criteria can be *approximately* satisfied by regularizing predictors (where the regularization crucially depends on the causal model of the features and label).

To extend the ideas of invariance to user decision predictions, the paper makes a key insight: the decision is affected by the causal model by which a user reasons about features to make a decision. Thus, a key idea in the paper is to parameterize such causal models of user reasoning. After formalizing a notion of CF invariance in the user decisions setting, the paper uses the idea of boundedly rational agents to instantiate a causal model of user decision making. In such a model, they articulate how paths in the graph can encourage predictors to exploit non-causal (i.e., spurious) information that doesn't remain stable across distributions. Then, the paper draws upon results from the work on CF invariance to propose regularizers to discourage predictors from using such non-causal information. Since the regularization scheme depends on the causal model, the paper distinguishes two different causal models of users that lead to different regularizers. They empirically study CF invariant perdictors of user decisions with recommendation data and simulated outcomes, finding evidence of robustness to distribution changes and substantiating that indeed, it is important to match regularizers appropriately with causal models.

**Questions:**

I described the main technical gaps I perceived above. I'll reiterate the points as clear questions for the authors here:

1. Can you clarify whether in definition 1? Does the invariance have to hold for every possible unit (i.e., this is truly counterfactual and would be impossible to verify without assumptions on the full SCM)? If so, why bother since the graph can't identify such a criteria anyway?

2. Can you clarify the motivating figure 2 model, and explain why the $e \rightarrow y$ link is being considered "spurious" or non-causal here? As explained in more detail above, invariant causal learning assumes that the environment has no causal effect on the target $y$ because if it did, environment-invariance isn't intuitively a sensible ask to begin with.

3. Can you clarify that my understanding of S3.3 was right, and clarify why learning a flexible predictor from $\bar{x}, x$ and $r$ (if $\bar{x}$ was observed) isn't the optimal thing to do, since we could just learning the (invariant) causal mechanisms directly?

4. Can you better justify how, in the anti-causal model, the user's decision $y$ could causally precede the perception of value?

5. Can you articulate what CI tests you're envisioning to distinguishing between anti-causal and causal user models (presumably, testing to see if $y$ is a collider for $x$ and $r$)?

6. Can you generally clarify proposition 2, perhaps with an example?

**Limitations:**

As discussed in the weaknesses above, I'd encourage the authors to spend more time discussing the challenges underpinning distinguishing between anti-causal and causal models. Moreover, both in exposition and empirically, it would be good to see more about how to practically deal with the issue of users in the data coming from a mix of these models.

**Strengths And Weaknesses:**

Strengths:
1. The paper takes a new approach to robust recommendations, making the novel connection that for invariant decision prediction, modeling the user decision making process is what is relevant. In this vein, using ideas from boundedly rational agents, the paper instantiates some causal models of users and articulates what paths in these models incentivize predictors to use spurious information. These connections are non-trivial to develop and can inspire extensions and new veins of work in the recommendation setting.
2. The empirical studies that involve recommender data with simulated outcomes can be considered another contribution in this paper. Such experimental evaluation could also provide new tasks and baselines for those looking to work on the problem of invariant recommendations.
3. The writing is generally clear and the authors do a good job or motivating technical ideas with examples and intuitions. The authors seem to have to covered related work well too.

Weaknesses:
1. In applying and extending ideas about CF invariance, there are some technical gaps and ambiguities throughout the paper that leave me doubting the soundness of some ideas. I'll enumerate:
+ the notion of CF invariance has always been a bit poorly formalized. Consider definition 1. Does the invariance have to hold for every possible unit (i.e., this is truly counterfactual and would be impossible to verify without assumptions on the full SCM)? Or, does $f_u(\cdot)$ have to be invariant for all values of x and r, i.e., $P(f_u(x(e), r(e) | X=x, R=r; \theta) = P(f_u(x(e'), r(e') | X=x, R=r; \theta) \forall x, r$? Or, do you want it to hold on average across interventions to $e$, i.e., $P(f_u(x(e), r(e); \theta) = P(f_u(x(e'), r(e'); \theta)$? Definition 1 isn't well defined enough to make these distinctions clear. Note that causal graphs alone could only identify versions 2 or 3 of this invariance.
+ In the same vein, if definition 1 is intended to be truly counterfactual (i.e., must be satisfied for every possible unit specified by the SCM), since the causal graph can never identify it anyway, why bother? Why not just pick a population-level (i.e., interventional as in version 2 or 3 above) version of invariance to satisfy?
+ The motivating causal model in figure 2 is a bit strange because in this model, the decision truly is causally affected by the environment. In this case, a predictor that satisfies CF invariance in Definition 1 would not capture the causal process of user decision making, which is often the underpinning of invariant causal learning. In fact, much of the work on invariant causal learning begins by assuming that the environment has no causal effect on the target $y$ because if it did, environment-invariance isn't intuitively a sensible ask to begin with. However, in this paper, the $e \rightarrow y$ motivation persists and is described as the main pathway by which predictors exploit spurious information. It's not non-causally spurious information in this case, though, so the motivation is quite confusing.
+ The notation in S3.1 and figure 2 tend to obfuscate the fact that what we want is for some of the *information* in $x$ and $r$ to be discarded. That is, we want to partition the information in those variables into a part that a desirable predictor $f_u(\cdot)$ can use and a part that $f_u(\cdot)$ should't use. For example, the work by Veitch et al. makes this information decomposition explicit, thereby allowing us to reason about what graphical relationships the "nice" part of $x, r$ have to other variables. By being more coarse-grained and only encoding $x$ and $r$, however, it's not easy to reason about what parts of the variables an invariant predictor ought to use, and what relationships the parts have to other variables in the graph.
+ In S3.2 on "rational users", the authors say that a rational user model is akin to assuming a causal graph in which there is no direct arrow from $e$ to $y$, which handicaps the ability to avoid spurious pathways. This statement is confusing. A causal model with no arrow from $e$ to $y$ seems like the best case scenario because any flexible predictor from $x$ and $r$ to $y$ would be able to capture the causal mechanisms by which $y$ is produced. Perhaps a fairer assessment here is that although rational users constitute a best case scenario, this is not a likely model of users.
+ In S3.3, it seems like if we had the function $v_u(x, \bar{x}, r)$ (which doesn't depend on the environment), then I could construct an invariant predictor. But, since $\bar{x}$ is latent, a predictor is incentivized to reconstruct $\bar{x}$ information using the observed variables $x$ and $r$. However, since $e$ leaves its imprint on $x$ and $r$ and also affects $\bar{x}$, a predictor might just memorize $e$-relevant information instead. However, the authors say that using $\bar{x}$ if it *was* observed would be bad because it could lead to overfitting -- this doesn't match my intuition because I'd have thought that observing $\bar{x}$ is a best-case scenario because then, all paths from $e$ to $y$ are blocked and any flexible predictor could capture the causal mechanisms governing $y$ from $x, \bar{x}$ and $r$.
+ In S4.1, the story of the anti-causal user doesn't make sense to me because $v_u(x, \bar{x}, r)$ sounds like reflects the user's perception of value -- how could the user's decision $y$ possibly causally precede the perception of value? Put differently, the notation may be using $v_u$ to conflate true and perceived value.
+ In Prop 1, $y$ and $e$ being confounded is not $e \rightarrow y$. The authors should clarify what they mean here.

2. The paper relegates an important limitation to just 1 sentence. In S4.1, after articulating how the choice of anti-causal versus causal matters for regularization, the authors state that ``this can be done using focused interventions (e.g., A/B testing), or for observational data, using simple conditional independence tests.'' Conducting A/B tests to determine a causal arrow direction for each user is impractical. It's important to spend more time, then, discussing the orientation of edges using CI tests and addressing all the limitations therein. Presumably the idea is to test for v-structures between $x$ and $r$, and $y$? This needs to be explained more directly since in general, causal structure learning won't orient all edges.

3. In a similar vein to above, the empirical studies focus on one or the other causal model (anti-causal vs causal), without addressing the challenge the authors themselves motivated, that typically, a training distribution consists of a mixture of these two causal model modalities. I would love to see this issue tackled more.

3. Section 4.2 could do with a figure that helps to illustrate the proposition and the reasoning about v-structures. This section is hard to follow especially because we don't have the notation to talk about the parts of information that a CF invariant predictor uses. That being said, this proposition seems to provide a novel insight beyond the existing work of CF invariance, so perhaps it would have been more interesting in this paper to dwell on the differences between skeptics and believers than between anti-causal and causal users.

---

> ### Author Response · Authors · 2022-08-02
> **Author response to reviewer HSTu (Part 3/3)**
>
> (4) *”Can you better justify how, in the anti-causal model, the user's decision y could causally precede the perception of value?”*
>
> The user’s decision y does not precede the perception of value. Please refer to line 275, where we describe anti-causal users as “believing that an item’s *value* causes its description”. This is not to be confused with the user’s decision, which is a function of her perceived value. An anti-causal edge x<-v appears if the user *believes* the information x presented to her by the system depends on the item’s value (the notation x<-y, which “folds” v->y into y (line 206), may indeed be confusing in this regard; we will clarify this). One scenario in which this can occur is when a user believes that the system can accurately estimate v - perhaps even better than the user herself can - and sets x and/or r accordingly. As an example, consider a restaurant recommendation service which reveals information about restaurants that matches (predicted) user tastes: e.g., a tempting menu item (for a restaurant that is predicted to match her tastes), or an overcrowded seating area and a long line (for a restaurant which does not).
>
> Note that a descriptive model of anti-causal user *decisions* requires being precise about the role and form of perceived values; these can be distinct from those described for the causal user in Eq. (1).
>
> (5) *”Can you articulate what CI tests you're envisioning to distinguishing between anti-causal and causal user models (presumably, testing to see if y is a collider for x and r)?”*
>
> Please see our detailed response to all reviewers on this subject in the general thread above.
>
> (6) *”Can you generally clarify proposition 2, perhaps with an example?”*
>
> On a high-level, Prop. 2 begins by stating that it is possible for two users of the same type but of different sub-types - namely skeptics vs. believers - to generate very similar observational data (the proposition itself portrays an extreme case where they can be identical, but the overall message is that observational data may conceal important variation in causal user perceptions). Next, consider that if we were to train a predictive model by minimizing the loss on data generated by one of these users, then we would expect this model to also have low loss for data generated by the other user (note that since both users are of the same type, Prop. 1 states that they require the same regularization, and so their objectives match). Hence, it would seem beneficial to pool both datasets together - since this would provide more data from the “same” distribution - and train a single predictive model. Indeed, if our goal was to optimize in-distribution performance, then this reasoning would be correct; however, the key statement of Prop. 2 is that for out-of-distribution performance, *it would be better to train different models*, one for each user, and independently. The conclusion carries over to populations of same-type users that consist of two sub-populations of different sub-types.
>
> Formally, in our proof we construct a data generating process that is entailed by the graph in Figure 6. We show that under certain settings for this DGP, if we sum out all variables other than $x,r,y$ (in our construction this means summing out a confounder $c$ and the environment variable $e$), we may arrive at the same joint distribution over $x,r,y$ for two users where one is a believer ($r{\rightarrow}x_{sp}$, for some features $x_{sp}$) and one is skeptic ($r{\leftarrow}x_{sp}$). But whereas for the believer, the optimal CI predictor takes both $r$ and $x_{ac}$ into account, this model is not CI for the skeptic user and hence they do not share optimal robust models.
>
> The proof is constructive and one may easily construct numerical examples, one such example is constructed for the simulation in section 5.3. Notice that in Table 1, when $\lambda=0$, the performance is the same for all user subclasses, regardless of the subgroup we trained on. Implying that the non-robust models learned via ERM are the same for both users. It is only when we learn a robust model (i.e., $\lambda > 0$) that we observe the benefits of learning a different model for each subclass.

---

> ### Author Response · Authors · 2022-08-02
> **Author response to reviewer HSTu (Part 2/3)**
>
> (2) *”Can you clarify the motivating figure 2 model, and explain why the $e{\rightarrow}y$ link is being considered "spurious" or non-causal here? As explained in more detail above, invariant causal learning assumes that the environment has no causal effect on the target y because if it did, environment-invariance isn't intuitively a sensible ask to begin with.”*
>
> In our example (which Fig. 2 illustrates), it is $\bar{x}$ that becomes spurious - not $e$. To see this, note that data is generated by a user model in which $\bar{x}$ is only indirectly related to $y$ (through integration). We apologize if this was unclear.
>
> Indeed, in some cases we present e as directly affecting $y$; however, this should *not* be taken to mean that we assume that all causal paths to $y$ originate in $e$. Fig. 2 is intended to depict the minimal structure necessary to convey our modeling ideas, and so presents a simplified generative process; the framework itself does not prevent $x$ or $r$ depending on some other variable that is unrelated to $e$, so that some elements of $x,r$ are invariant. This is made explicit in Sec 5.2 (see Fig. 6) and in Appendix A (see Fig. 7).
>
> (3) *”Can you clarify that my understanding of S3.3 was right, and clarify why learning a flexible predictor from $\bar{x}$, $x$ and $r$ (if $\bar{x}$ was observed) isn't the optimal thing to do, since we could just learning the (invariant) causal mechanisms directly?”*
>
> To the best of our understanding, we believe your intuition here is wrong. To see this, recall the important distinction that while the system observes $\bar{x}$, the user - whose decisions determine $y$ - does *not* (lines 131; 196). The decision rule (Eq. (1)) accounts for (user-side) uncertainty regarding $\bar{x}$ by summing over possible assignments to $\bar{x}$, and so the true mechanism which generates $y$ relies on distributional information regarding $\bar{x}$ - *not* on specific instantiations. This means that the generation of $y$ does not depend on $\bar{x}$ as input.
>
> However, if the system learns functions of the form $f(x,\bar{x},r)$ from inputs of the form $(x,\bar{x},r,y)$, then learning will likely utilize variation in $\bar{x}$ to explain y, if they correlate (nothing in the objective discourages this). This would mean that the learned $f(x,\bar{x},r,y)$ will vary with $\bar{x}$ -  whereas the true generating process of $y$ does not - and in a way that fits the training environments; hence, such a model is unlikely to generalize well to new environments.
>
> A subtle point is that if the system has access to data which includes instances of $\bar{x}$, then it also has access to the (empirical) distribution of $\bar{x}$. This means that, at least in principle, the system can utilize this distributional information in learning. However, nothing in the learning objective encourages learning in this particular way.

---

> ### Author Response · Authors · 2022-08-02
> **Author response to reviewer HSTu (Part 1/3)**
>
> Thank you for your detailed review. We appreciate your careful attention to detail, and will be sure to revise the paper in a way that makes these details readily available to readers who are interested.
> In your review, you express concern for the “soundness” of our approach. We fail to see why - nor do we see any comment in the review that points to a concrete source of unsoundness. To the best of our understanding:
> * Q1 and Q6 regard *clarification*. In our response we both clarify and provide additional details.
> * Q2, Q3, and Q4 reflect what we believe are *misunderstandings* by the reviewer.
> * Q5 presents what you perceive as a limitation. We respectfully disagree, and present an alternative viewpoint.
>
> Having addressed these issues - and given our clarification in the general thread above as to what we consider to be our main focus and contributions - we are hopeful that you would be willing to favorably reconsider your evaluation.
>
> **Detailed response:**
>
> (1) *”Can you clarify whether in definition 1? Does the invariance have to hold for every possible unit (i.e., this is truly counterfactual and would be impossible to verify without assumptions on the full SCM)? If so, why bother since the graph can't identify such a criteria anyway?”*
>
> Technically, and in its most general form, the result from Veitch et. al. (2021) on which Prop. 1 is based requires unit-level invariance. As Veitch et. al. show, population-level invariance (which regularization promotes) can be implied from unit-level invariance under further fine-grained assumptions on the graph, and in particular, using variables which are not affected by e (denoted $x^{\perp}_e$, see proof in Appendix A). In our work, we make this assumption explicit where necessary (e.g., Sec. 5.3; see Fig. 6), but intentionally abstract away from it when it is not. For example, Sec. 3 focuses on the decision-making aspects of modeling; hence, Fig. 3 does not explicitly show an $x^{\perp}_e$ variable because it is not essential to understanding the relation between user beliefs and the system’s modeling choice. At the same time, there is nothing preventing a modeler from including such a variable - the example holds just as well with it.
>
> However, we agree with your point on the importance of understanding the connection between structure and Prop. 1, and will clarify the importance of being precise about relational assumptions when modeling in concrete tasks.
>
> In regards to whether population-level invariance is helpful: as Veitch et al. comment, population-level invariance is a necessary (though not sufficient) condition which, under observational data, is likely “the closest proxy for counterfactual invariance we can hope for” (see section “Gap to Counterfactual Invariance” in their paper). Veitch et al. support this claim with experiments; we believe we do so as well, especially in our new experiment on mixed populations (see general response above). Whether this claim holds more broadly (or not) will hopefully be determined as further evidence amounts, and as more useful ways of achieving user-level invariance are discovered, by future works in this young but growing line of research (e.g., see the recent work in https://arxiv.org/abs/2207.09768).

---

> ### Comment · Reviewer_HSTu · 2022-08-04
> **Thanks for your detailed response**
>
> Dear authors,
>
> Thanks for taking the time to write a detailed response and answer all of my questions. I want to start by clarifying something you mentioned in your reply: you didn't see why I would doubt the soundness of your approach. I want to clarify that, as I wrote in my detailed comments, in the submitted manuscript, there was enough confusing notation that I felt there was a gap between what was written in plain English and what appears in the graphs or the math.
>
> Your answers gave me some clarity but I'd like to still drill down into some points.
>
> 1. Re: answer to: ``(2) Can you clarify the motivating figure 2 model?"
>
> Yes, I understand that it's the presence of $\bar{x}$ that creates spuriousness: as you write in section 3.3, a classifier that uses only $x$ and $r$ could use $e$ information contained in $x$ and $r$ to compensate for not knowing $\bar{x}$. However, I don't agree that figure 2 provides an easily understandable minimal example. Figure 2 causes confusion: it suggests that the environment truly causes the decision $y$, in which case the CF invariance to $e$ in definition 1 doesn't really make sense. Since Figure 2 is used to explain the problem formulation, I think it makes sense to go straight to something like Figure 3 left, which is makes the source of spuriousness *much* clearer.
>
> 2. Re: answer to (3) "If $\bar{x}$ is actually observed, why isn't learning a flexible predictor from $x, \bar{x}$ and $r$ the optimal thing to do?"
>
> I don't really follow your intuitions. To clarify my thinking here, put simply, in figure 3 (left), $x, \bar{x}$ and $r$ block all paths from $e$ to $y$. Thus, if we had $\bar{x}$, it seems like using it alongside $x$ and $r$ to predict would be correct. You write in S3.3 that "the reliance of users on 𝑥̄ for producing 𝑦 [...] means that, effectively, such an edge exists." I don't understand how to reconcile this with your point in the reply that "This means that the generation of $y$ does not depend on $\bar{x}$ as input."
>
> Could you provide some more intuitions or clarity for how to understand your answer, especially in terms of Figure 3 left?
>
> 3.  Re: answer to "4) Can you better justify how, in the anti-causal model, the user's decision y could causally precede the perception of value?”
>
> Yes, I understand that perceived value could cause $x$ and $r$. However, writing $y \rightarrow x,r$ when $y$ was used to denote the decision is indeed confusing -- it conflates the decision with the true value. Your reply helps and some rewriting of this section will help. To substantiate my original comment around soundness, this notation is one example that had me feeling that there was a technical gap.
>
> 4. Re: Proposition 2
>
> Thanks for your explanation. I appreciate the time you put into adding extra empirical studies to the implications of the result.

---

> > ### Author Response · Authors · 2022-08-09
> > **Clarifying remarks**
> >
> > Thank you again for your detailed inquiries and thoughtful comments. We appreciate your evident and highly non-trivial reviewing efforts.
> >
> > 1. *“I don't agree that figure 2 provides an easily understandable minimal example.”*
> >
> > We would like to begin with an additional clarification. Since our work lies at the intersection of several disciplines, our paper is written in a way that is intended to be read by multiple audiences, and our choices regarding structure, wording, notation, and illustrations reflect this. In particular, our choice of graphical examples was designed to aid readers who are not necessarily fluent in invariant learning. For these readers, our reasoning was that graphs should:
> >
> > (i) convey the basic structure around our user model,
> >
> > (ii) gradually progress from simple to complex, and
> >
> > (iii) first focus on where and how spuriousness can arise (e.g., user behavior), and only then on how invariance can be achieved (regularization and additional structure), these being in line with our focus in Sec. 3 and Sec. 4, respectively.
> >
> > In these three senses - and at least in our minds - Fig. 2 is “minimal”. Our belief was that Fig. 2 would help readers (who are not accustomed to examining graphs for possible handles on invariance) think about spuriousness in a way that sets the ground for the behavioral model and Fig. 3 in Sec. 3. However, as Fig. 2 is more confusing than helpful, **we gladly remove it altogether**.
> >
> > 2. *“if we had $\bar{x}$, it seems like using it alongside x and r to predict would be correct.”*
> >
> > In line with your views - we would first like to clarify the math behind the model, and then discuss possible misunderstandings regarding the illustration in Fig. 3.
> >
> > **The role of $\bar{x}$ in prediction:**
> > Begin by considering - irrespective of Fig. 3 - a user choosing via Eq. (1). Fix e, and consider a distribution $D^e_{X,\bar{X},R}$. What would describe the generating mechanism of y? Eq. (1) states that y (which is a deterministic function of $\tilde{v}$) depends on x,r, and e - not $\bar{x}$. In other words - if we sample only $x,r \sim D^e_{XR}$, then this provides sufficient information for determining $y=\tilde{v}(x,r|e)$; had we then sampled $\bar{x} \sim D^e_{\bar{X}|x,r}$, this would not have any effect on y, nor would such an instance of $\bar{x}$ contain any additional information regarding the jointly-sampled y beyond that in x,r.
> > Consider now a system that aims to learn $y$ for such a user. The system observes samples $(x,\bar{x},r,y)$, and so it may seem appealing to learn a mapping $\hat{y}=f(x,\bar{x},r,y)$. However, $y$ does *not* depend on $\bar{x}$. Hence, relying on $\bar{x}$ in $f$ would *not* be optimal, since learning may (wrongly) use correlations between $\bar{x}$ and $y$. Furthermore, assuming (incorrectly) that conditioning on $x,\bar{x},r$ blocks all paths from $e$ to $y$ would (wrongly) imply that `naively’ training $f$ using e.g. standard ERM should suffice for learning a robust predictor. But in effect, it would not.
> >
> > **Graphically illustrating dual perspectives:**
> > Now, return to Fig. 3 (left). Importantly, the diagram is *not a causal graph*. To see this, note that (i) in line with the above, a graph depicting the DGP of y would *not* include an $\bar{x}$ variable, and (ii) the $\bar{x}$ variable in the diagram is *integrated over* - an operation which is not depicted by causal graphs.
> > What the diagram does convey is a *model of user beliefs* (in which $\bar{x}$ does play an active part), as is common in choice modeling (and would likely be evident to a reader from this field). What we believe may have caused confusion is the attempt to connect this model to the pursuing causal graph - which is obtained via integration and choice - *in the same diagram*, and in a way that simultaneously presents the perspectives of both the system (which observes instances of $\bar{x}$) and the user (which integrates over its possible values and w.r.t. personal beliefs).
> > Given your helpful feedback, **we have now decomposed Fig. 3 (left) into two distinct components: a model of user beliefs, and the resulting DGP.** We have also revised the caption to provide more detail on differences and connections, and will continue to revise the text in a way that better conveys these subtle points. We have also made it clearer that Fig. 3 is intended to show only how spuriousness can arise, and not how to treat it (which, as you note, requires additional structure), and use it to set the stage for Sec. 4 on invariant learning.
> > We are hopeful that our reasoning regarding the role of $\bar{x}$ in prediction is now made clear (at least in terms of how spuriousness can be inadvertently missed), and is better conveyed in both writing and illustrations.
> >
> >
> > 3. “Writing y→x,r  when y was used to denote the decision is indeed confusing.”
> >
> > We agree, and thank you for pointing this out. We will make sure to decouple y from v when needed in the revised version of the paper.

---

> > > ### Comment · Reviewer_HSTu · 2022-08-09
> > > **really helpful reply**
> > >
> > > I really appreciate the time you've taken to engage with all my questions and comments. I found your answers helpful for my understanding. I also reviewed some of the changes you've made to the manuscript and I agree with you that your changes to Figure 3 really help. As such, I've revised my score upwards.

---

### Official Review · Reviewer_zV27 · 2022-07-11

**Rating:** 6
**Confidence:** 1
**Soundness:** 2 fair
**Presentation:** 3 good
**Contribution:** 3 good

**Summary:**

This paper proposes a learning framework that is robust to environment changes to predict the relevance of items to users. The key idea is to model users as reasoning about decisions through a causal graph and show how minimal information regarding the graph can be used to contend with distributional changes.

**Questions:**

It seems the prior knowledge regarding users' causal beliefs (causal or anti-causal) needs to be known ahead for modeling. How can such information be obtained in real-world tasks?

**Strengths And Weaknesses:**

The proposed framework is novel and the paper is fairly well written, and experimental results support the paper's statement that adopting users' causal beliefs results in better prediction performances.

However, there are no comparisons with other exciting works. According to section 2 related works, there exist, several models that adopt causal inference techniques for recommendation tasks, the paper would be more sound if the prediction performances are compared against those works.

---

> ### Author Response · Authors · 2022-08-02
> **Author response to reviewer zV27**
>
> Thank you for your reviewing efforts; although our paper is likely not in your main area of expertise, we nonetheless hope it served as an interesting and pleasant read.
>
> (1) *No comparisons with other exciting works (e.g., those in Sec. 2):*
> While the papers surveyed in Sec. 2 relate to causal inference in the general setting of recommendation, their focus is quite different than ours: [29] focus on inference, and in particular aim to correct for exposure bias; [56] present a method for “deconfounding” possible confounders; [7] seek to find a policy that maximizes ITE, rather than predicting; [58] propose to leverage item popularity as a confounder; [55] study causes for bias amplification; etc. Hence, we do not see how they can be applied to our setting.
>
> (2) *It seems the prior knowledge regarding users' causal beliefs needs to be known ahead for modeling. How can such information be obtained in real-world tasks?*
> Please see our detailed response to all reviewers on this subject in the thread above.

---

### Official Review · Reviewer_EVAJ · 2022-07-15

**Rating:** 7
**Confidence:** 4
**Soundness:** 4 excellent
**Presentation:** 3 good
**Contribution:** 3 good

**Summary:**

This paper studies the problem of predicting item relevance to users, in a way that is robust to changes in the data distribution. It is a particular case of tackling the bigger challenge of out of distribution generalization - here different environments are represented by different users' beliefs. The paper gives a strategy for trustworthy relevance prediction which does not require full information of the underlying causal graph. It only requires one to know whether the graph is causal or anti-causal, and for each of these scenarios an appropriate regularization scheme is chosen.

**Questions:**

My only question is the same as Point number 1 in weaknesses.

**Limitations:**

The authors have mentioned limitations of their method both in the discussion section as well as in the description of their experimental results.

**Strengths And Weaknesses:**

Strengths:

1. The paper is very well-written and the motivation and setup are very clear. I enjoyed reading this work.
2. The learning algorithm proposed requires minimal knowledge of the causal graph (i.e. whether it corresponds to (a) causal and (b) anti-causal with two possible subclasses - believer or skeptic; this latter sub-distinction is very nice and quite realistic for modeling user behavior).
3. The connection to economics and user behavior psychology by modeling the response y via an expected utility function is interesting.
4. This model also accounts for the case in which some intrinsic information about the product (encoded in x) is missing.
5. The algorithm is clear and the experiments are thorough, accounting for all possible scenarios.


Weaknesses:

1. While I intuitively understand it, it is still unclear to me how the MMD and the CORAL regularization methods satisfy the respective necessary conditional independence requirements needed for preventing spurious correlations. Similarly, it is not completely mathematically clear to me why the conditional independencies described in lines 216-217 are satisfied.
2. Inferring the type of causal and non-causal graph is still non-trivial. The authors do mention some approaches for this in lines 294-295, i.e. A/B testing or conditional independence testing from observational data, but the paper could significantly benefit from a more developed discussion of possible ways to infer the graph in practice.

---

> ### Author Response · Authors · 2022-08-02
> **Author response to reviewer EVAJ**
>
> Thank you for your encouraging review, please see our response to your questions below.
>
> (1) *How do the MMD and the CORAL regularization methods satisfy the respective necessary conditional independence requirements needed for preventing spurious correlations?*
> Consider two environments, $e_1$ and $e_2$. For marginal MMD, note that the penalty constitutes a metric between the distributions of representations in each environment, $D^{e_1}(\phi(x,r))$ and $D^{e_2}(\phi(x,r))$. MMD encourages these distributions to be similar; hence, when the penalty is low, we can expect to have $P(\phi(x,r) \mid E=e) \approx P(\phi(x,r))$ for $e \in {e_1, e_2}$, i.e., that the probability of observing a certain representation is independent of the environment from which it was sampled. This means that for a predictive model that operates on these representations, the desired independence (approximately) holds.
> For conditional MMD, stratifying over y and then using the same reasoning gives the necessary conditional independence.
> As for CORAL, while it is not a proper metric between distributions, it does match certain properties of the distributional (i.e. covariance matrices), and is hence used in practice as a proxy for distribution matching (and often works better than MMD).
>
>
> (2) *Not completely mathematically clear why the conditional independencies described in lines 216-217 are satisfied:*
> The conditional independencies in lines 216-217 merely exemplify how different beliefs can lead to different factorizations, irrespective of the (possibly different) conditional independencies needed for robust learning.

---

### Author Response · Authors · 2022-08-02
**Author response: general notes to all reviewers**

We thank all reviewers for their efforts and valuable comments, which we will implement in our next revision of the paper, hopefully in the coming week. In this thread we discuss two themes that came up in more than one review. All other points are addressed in individual responses below.

(1) **Focus and contribution:**
Our key goal in the paper is to show how accounting for users’ causal perception can benefit robust learning. In accordance, our primary focus is on *modeling*, and we were happy to learn that all reviewers are appreciative of our efforts in this regard. To this end, we rely on recent advances in invariant learning, and we believe our result in Prop. 2 is a humble contribution to this growing literature. However, we use the framework of counterfactual invariance mostly as the best currently-available testbed for exploring our ideas regarding causal user modeling. As we state in the paper, we view our main contribution as being *conceptual* - our paper is about *making an intriguing and non-trivial connection between three distinct fields*. As to the connection to counterfactually-invariant learning, we believe our framework is flexible enough to remain relevant and applicable in future iterations of this young domain, as it evolves.

(2) **What we mean by “graph knowledge”:**
All reviewers requested, in one way or another, that we further elaborate on possible means to infer user graphs. Since our work considers the question “how can knowledge regarding users’ causal beliefs be useful?”, naturally we place emphasis mostly on what to do with such knowledge - rather than how to obtain it. Nonetheless, as per your request, we will gladly provide more details about graph inference in our next revision of the paper.

At the same time, **we would like to make an important clarification in this regard**. While there exist disciplined ways to infer graphs or orient certain edges - either from observational data (i.e., from the extensive literature on causal graph discovery) or through experimentation (i.e., using the established practice of A/B invariance testing) - an important point we wish to make is that our approach does not require, per se, precise and verified graph knowledge on each individual user. Rather, we envision our framework as applicable in cases where there is *good prior knowledge* regarding users’ causal beliefs, possibly even at the population level. For example, our beers experiment is designed to capture a setting where users are likely to be casual; our fashion experiment targets a setting where users are likely to be anti-causal.

The use of domain knowledge to determine the learning objective (e.g., how to regularize) is common practice in machine learning. For example, practitioners apply L1 regularization when they *believe* the true underlying model is likely to be sparse - not because some formal verification test guarantees this (and not for a lack of theory on sparse learning, on which there is plenty). Likewise, and relevant to our case, when there is good reason to believe that most users are of one “causal” type or another, we view our results as providing guidance as to how to devise the learning objective: Prop. 1 suggests what regularization to apply, Prop. 2 suggests treating different sub-populations independently.

To further ground this perspective, **we have added to the paper an additional experiment** on mixed populations. Focusing on believers vs. skeptics, we demonstrate how even a rough guess of how to partition the population (and learning “as if” each noisy sub-population was actually pure) still provides significant improvement in predictive outcomes. Our current revision includes these results and key setup details (Sec. 5.4), and we will continue to update with more details in the coming week.

Overall, we hope our refined perspective and new results help shed a different light on how our framework can be interpreted.

---

### Meta-Review · Area_Chair_EKUN · 2022-08-27

**Recommendation:** Accept
**Confidence:** Certain

**Metareview:**

This paper studies user-item relevance prediction and proposes a novel learning framework that is robust to distributional shifts in observed user-item attributes. All the reviewers appreciated the significance of the problem, the novelty of the solution, and the thorough empirical evaluation. The reviewers were confused by the exposition in some places, and the authors comprehensively addressed the questions during the feedback phase. Please include the extensive clarifying discussions with the reviewers in the revised paper, which will likely be of interest to the community.

**Award:**

No

---

### Decision · Program_Chairs · 2022-09-14

Accept